# Immune Stimulation with Imiquimod to Best Face SARS-CoV-2 Infection and Prevent Long COVID

**DOI:** 10.3390/ijms25147661

**Published:** 2024-07-12

**Authors:** Ursino Pacheco-García, Elvira Varela-López, Jeanet Serafín-López

**Affiliations:** 1Department of Cardio-Renal Pathophysiology, Instituto Nacional de Cardiología “Ignacio Chávez”, Mexico City 14080, Mexico; 2Laboratory of Translational Medicine, Instituto Nacional de Cardiología “Ignacio Chávez”, Mexico City 14080, Mexico; elvira.varela@cardiologia.org.mx; 3Department of Immunology, Escuela Nacional de Ciencias Biológicas (ENCB), Instituto Politécnico Nacional (IPN), Mexico City 11340, Mexico; jserafinl@ipn.mx

**Keywords:** COVID-19, Long COVID, SARS-CoV-2 immunization, imiquimod immunostimulation

## Abstract

Through widespread immunization against SARS-CoV-2 *prior* to or *post*-infection, a substantial segment of the global population has acquired both humoral and cellular immunity, and there has been a notable reduction in the incidence of severe and fatal cases linked to this virus and accelerated recovery times for those infected. Nonetheless, a significant demographic, comprising around 20% to 30% of the adult population, remains unimmunized due to diverse factors. Furthermore, alongside those recovered from the infection, there is a subset of the population experiencing persistent symptoms referred to as Long COVID. This condition is more prevalent among individuals with underlying health conditions and immune system impairments. Some Long COVID pathologies stem from direct damage inflicted by the viral infection, whereas others arise from inadequate immune system control over the infection or suboptimal immunoregulation. There are differences in the serum cytokines and miRNA profiles between infected individuals who develop severe COVID-19 or Long COVID and those who control adequately the infection. This review delves into the advantages and constraints associated with employing imiquimod in human subjects to enhance the immune response during SARS-CoV-2 immunization. Restoration of the immune system can modify it towards a profile of non-susceptibility to SARS-CoV-2. An adequate immune system has the potential to curb viral propagation, mitigate symptoms, and ameliorate the severe consequences of the infection.

## 1. Introduction

Nearly four and half years after the start of the pandemic triggered by the SARS-CoV-2 coronavirus [1,2], the tally of reported infections has exceeded 775 million, with the death toll surpassing 7,051,000 [3]; this has inflicted profound health and economic distress upon nations, communities, and individuals alike. Moreover, pockets of the population in various countries remain unvaccinated due to diverse factors [4]. While the containment of infections is generally deemed to be improving, instances of both initial infections and reinfections persist. A proficient immune system controls the viral infection in a short time without significant complications, a trait fortunately possessed by the majority of the population [5]. However, some population groups are more susceptible to developing severe COVID and post-COVID symptoms, including older adults, male adults, people with diabetes, those with hypertension, lung damage, or obesity, and individuals with immunodeficiencies, mainly. These groups of people have a reduced or dysregulated immune function that can lead to exaggerated inflammatory responses and inadequate protection against infections. Therefore, it is crucial to recommend vaccinations against seasonal influenza and other infections. The significance of a robust and properly functioning immune system became evident during the SARS-CoV-2 pandemic. Approximately 10% of individuals without prior immunity who contracted SARS-CoV-2 experienced severe COVID-19 symptoms [5,6,7], and at least 3% of these cases resulted in fatal outcomes.

In response to a viral infection, the proficient innate immune response mechanisms are the first to react by suppressing viral replication and, in some cases, eradicating the virus. This initial response hampers viral replication, affording the adaptive immune system the necessary time to mature and effectively manage the infection [5]. Innate and adaptive immune responses have the ability to control and eliminate the virus; the main innate response to the virus consists of the interferons, while the adaptive immunity responds with the development of antibodies and specific cytotoxic T cells. Both systems work properly when they are well regulated.

Senescence in both innate and adaptive immune cells increases with age, degenerating 59 immune system functions and their regulation [8,9,10,11]. Monocytes and DCs have impaired activation in older people and produce fewer antiviral substances [12,13,14,15], while inflammatory mediators are increased [16,17,18,19,20,21], senescent B and NK cell functions are altered [22,23,24], and there is a decrease in T cell receptor diversity, naive T cell deficiency, and alterations in miRNAs from immune cells [8,9,10,11]. Immunosenescence is also increased in metabolic syndrome [24]. So, immunosenescence related to metabolic syndrome is aggregated with immunosenescence related to age in obeses and diabetics adult.

This decline in immunity is increased with age and with the presence of comorbidities such as diabetes, obesity, hypertension, pulmonary damage, and others [8,9,10,11]. Individuals with these comorbidities have increased vulnerability to SARS-CoV-2 [11,25,26,27,28]. One such factor is their elevated expression of the ACE2 protein, which serves as the cellular receptor for the virus, facilitating a higher rate of cellular infection and consequently a greater viral load [29,30]. Additionally, the metabolic stress experienced by the immune system cells can disrupt their normal cellular function, and the persistent, systemic low-level inflammation seen in individuals with metabolic syndrome increases their susceptibility to developing a cytokine storm [30,31,32,33,34] when exposed to more significant triggers, such as infectious agents.

## 2. SARS-CoV-2 Infection and the Development of COVID-19 and Long COVID

SARS-CoV-2 infection manifests differently depending on individual factors like the initial viral load and immune system status. Infected individuals may (a) remain asymptomatic; (b) experience mild symptoms akin to the flu, including nasal and throat discomfort, fever, and a slight cough, often with minimal impact and fast resolution; (c) develop moderate symptoms characterized by prolonged high fever and cough, typically resolving within 15 days; (d) present more severe clinical manifestations, featuring a dry cough, high fever, pneumonia, hypoxia, respiratory distress, diarrhea, muscle pain, lung damage that can progress to acute respiratory distress syndrome (ARDS), multiple organ damage, coma, and even fatality. Common additional symptoms may include diarrhea and muscle pain [34,35,36,37,38,39,40,41].

Moderate to severe COVID-19 tends to affect elderly individuals, as well as those with obesity, diabetes, or hypertension. All of them may experience immune deregulation due to chronic inflammation. Additionally, individuals with pre-existing respiratory conditions and compromised lung function are at a higher risk of severe respiratory symptoms due to SARS-CoV-2. Certain genetic factors also play a role in susceptibility [27,28,35,42,43,44,45]. Furthermore, a significant proportion of individuals who experienced severe COVID-19 developed a condition known as Long COVID (LC), characterized by a range of symptoms, including cardiovascular, inflammatory, renal, endocrine, and neurological manifestations, which can vary in severity and some of which can be disabling. These symptoms can persist for several months [37,38,46,47]. Individuals of advanced age with diabetes, obesity, or hypertension often exhibit a chronic, low-intensity systemic inflammation profile characterized by elevated proinflammatory interleukin levels, such as IL-6 [27,46,47,48,49,50,51,52]. They also tend to have higher expression of the ACE2 receptor and lower levels of type 1 interferons (IFNs) [34,48,49]. This combination predisposes them to an exaggerated inflammatory response mediated by cytokines at the onset of infections, along with a deficient specific humoral and cellular immune response, increasing their susceptibility to developing severe COVID-19 and LC. A common but not exclusive characteristic found in individuals who develop both severe COVID and LC is the presence of the FoxP4 locus. Four variants of the FOXP4 gene have been found in individuals who developed LC [53,54]. Expression of these variants was found in both lung cells and immune cells in the lung tissue of individuals who developed LC, whether they had severe or moderate COVID-19. These FOXP4 variants explain at least part of the development of LC; it is due to lung dysfunction, such as fatigue and chronic respiratory problems due to lung damage caused by COVID-19, since other chronic symptoms manifest in LC due to the involvement of other damaged organs. Variants of other genes associated with the development of severe COVID-19 and LC have been reported in other population studies [53,54]. This locus is also associated with lung dysfunction and various types of cancer. On the other hand, differences in the serum miRNA profiles between individuals who develop severe COVID-19 or LC and those who recover adequately have been reported [55,56,57]. Differences in the levels of proinflammatory cytokines, acute-phase proteins, and proteins secreted by macrophages, as well as high levels of anti-inflammatory osmolites (taurine and hypaphorin), have been found in those who developed LC compared to those who recovered adequately from the infection [58].

Commonly, the longer the duration and severity of the infection, the greater the damage caused to different organs and tissues, with a prolonged convalescence and the possibility of developing LC [59,60]. Rosichini M et al. demonstrate that infection of the thymus by SARS-CoV-2 causes damage to this organ and induces low production and maturation of T lymphocytes [61], which makes these individuals more susceptible to infections by other microorganisms or to the reactivation of latent infections such as those caused by viruses belonging to the herpes family, tuberculosis, or cytomegalovirus.

Besides Zheng YB et al. report that a percentage of children with severe COVID-19 develop prolonged multisystem COVID-19, especially children who have had at least one previous associated illness [62]; in a more recent study, Franco et al. report that individuals recovered from SARS-CoV-2 infection have a reduction in quality of life and an increase in the risk of developing at least two persistent symptoms later, in addition to metabolic, cardiovascular, neurological, respiratory, and hematological incidents after a diagnosis of SARS-CoV-2 infection [63]. Previous comorbidities are conditions that can facilitate the development of an impaired immune system and susceptibility to infections such as SARS-CoV-2 with a high risk of developing severe COVID-19, multisystemic infection, or LC, which is why it is necessary to stimulate an impaired immune system in order to develop an efficient immune response that quickly and efficiently controls the infection.

## 3. Impaired Immune Response against SARS-CoV-2

In individuals with impaired innate immunity, SARS-CoV-2 infection disrupts the interferon system’s ability to control viral replication. Once a high viral load is established, it can lead to the development of acute respiratory distress syndrome (ARDS) due to lung damage resulting from the cytokine storm. This condition is characterized by severe symptoms, including high fever, respiratory distress, hypercoagulation, and multiple organ damage. These complications can lead to a prolonged recovery period or, in some cases, a fatal outcome [59,64,65]. At the onset of coronavirus infection, these individuals exhibit a distorted and uncontrolled non-specific response, accompanied by a low-level, inefficient, or inadequate specific immune response [31,32,33,59,63,64,65,66,67,68,69]. Approximately 10% of survivors of severe infections go on to experience LC [27,28,70,71]. On the other side, vaccines administered to individuals with a deregulated immune system often result in an ineffective, low-level, and shorter-lasting immune response [5,70,71]. It becomes essential to bolster immune efficiency in individuals with these immune characteristics; this can be achieved through various treatments, including implementing immunostimulants, regular exercise, healthy weight management, a balanced diet, inhibition of inflammatory processes, and the use of antioxidants to reduce both intra- and extracellular oxidative stress [70,71,72,73,74,75,76,77,78,79,80,81], among other strategies. In this review, we analyze an immunostimulatory treatment regimen utilizing imiquimod (IMQ), which has the potential to positively modify and enhance the immune response in individuals with immune deregulation. Such an approach aims to facilitate the development of an efficient immune state for better control of SARS-CoV-2 infection and potentially mitigate or eliminate the adverse effects of LC.

## 4. TLRs in SARS-CoV-2 Infection

The presence of the SARS-CoV-2 virus and its components induce pathogen-associated molecular patterns (PAMPs) that are recognized by endosomal and cytosolic host cells’ pattern recognition receptors (PRRs), while damaged cells show damage-associated molecular patterns (DAMPs). These PRRs are sensors and effector molecules of the innate immune system, including Toll-like receptors (TLRs), retinoic acid-inducible gene-I (RIG-I)-like receptors (RLRs), nucleotide-binding oligomerization domain (NOD)-like receptors (NLRs), cyclic GMP-AMP synthase (cGAS), C-type lectin receptors (CLRs), absent in melanoma 2 (AIM2)-like receptors (ALRs), and stimulator of interferon genes (STING). PRRs´s activation induces the inflammatory signaling pathways and the immune responses that lead to the generation of IFNs, cytokine production, and apoptosis induction to eliminate viral replication and stop viral spread [82,83,84,85,86]. A deregulated innate immune system can lead to excessive immune activation with systemic inflammation and severe disease. In response to SARS-CoV-2, TLRs, RLRs, and NLRs induce cytokine production before activating their signaling pathways. 

Innate immune cells, including macrophages, monocytes, dendritic cells, neutrophils, innate lymphoid cells (ILCs), and natural killer (NK) cells, have at least five primary PRR families, which include TLRs, RLRs, NLRs, CLRs, and ALRs. PAMPs or DAMPs are detected by these PRRs signaling in innate immune cells [82].

TLRs are type I transmembrane (TM) proteins with three domains: an extracellular domain or ectodomain, a single-pass TM, and a cytoplasmic TIR downstream-signaling domain. TLRs are expressed in cellular membranes as homodimers or as heterodimers when encountering PAMPs or DAMPs, with the aid of adaptor proteins, initiating downstream signal transduction, inducing the expression of interferons and inflammatory cytokines (IFNs) [87,88]. 

There are ten human TLRs (TLR1 to TLR10). TLR1, TLR2, TLR5, TLR6, TLR7, TLR8, TLR9, and TLR10, which transduce signals via the adaptor molecule MyD88; TLR4 can bind and signal through either MyD88 or TRIF (Toll/interleukin-1 receptor/resistance protein (TIR) domain-containing adaptor-inducing interferon (IFN)-β) [89] and TLR3 only through TRIF. 

Innate immune cell populations of the human respiratory tract express TLRs, with NK cells expressing TLR3 and macrophages expressing TLR4 [89]. TLRs are expressed by other cells in airway tissues, too [90,91,92,93,94,95], including the epithelial cells, which are susceptible to SARS-CoV-2 infection. Viral RNAs activate the innate immune system through TLR signaling, activated MyD88 initiates NF-kB, MAOK, and interferon (IFN) regulatory factor (IRF) signaling, and its nuclear translocation of these molecules triggers inflammatory cytokine transcription, including TNF, IL-6, and IL-1 and other sensors such as NLRP3, and the production of type I IFNs and IFN-stimulated genes (ISGs).

Signaling via TRIF also activates type I IFN production and that of several transcription factors dependent on TLR4 and TLR3, including some with antiviral activity [89]. TLR7/8 recognizes RNA from SARS-CoV-2 and antiphospholipid antibodies (aPLs). These aPLs are increased in patients with severe COVID-19 [96,97]. TLR8 activated by aPLs induces an increment in type I IFNs in antiphospholipid syndrome (APS) [98], and this mechanism can participate in the aberrant type I IFN signaling and thrombo-inflammation present in severe COVID-19 patients [99].

Six TLRs recognize the presence of SARS-CoV-2 (TLR2, TLR3, TLR4, TLR7, TLR8, and TLR9), of which TLR2, TLR4, and TLR9 favor the pathogenicity of the virus, while TLR3, TLR7, and TLR8 favor control of the infection and resolution of the disease [100].

TLR7 is abundantly expressed in immune cells, including monocytes, macrophages, myeloid DCs, plasmacytoid DCs [101,102,103], T cells [101], B cells, and endothelial cells [104]. TLR8 is expressed in monocytes, macrophages, myeloid DCs [105], plasmacytoid DCs, T cells [101], and B cells [106].

Human TLR8 (hTLR8) is highly expressed in cells of the myeloid lineage, including neutrophils, and is not expressed in pDCs or B cells [107]. hTLR8 is activated by bacteria and bacterial RNA of various sources [108,109]. TLR7 and TLR8 share a similar mode of action [107], with two distinct sets of ligand-binding locations available [109,110,111]. 

Severe and fatal COVID-19 is associated with a lower expression of TLR7 and TLR8 compared to that in survivors [112,113,114].

Different TLR7 variant polymorphisms are present in a minority of the total population, and some of these variants have lost their function due to mutations in TLR7; this may drive severe COVID-19 in males because TLR7 is located on the X chromosome [112]. A study recently showed that macrophage-activation syndrome associated with missense mutations is due not only to TLR7 but also to TLR8; this is important particularly in males [115] since TLR8 is also located on the X chromosome [112]. A percentage of females are homozygous for some alleles; many can compensate for mutations with the other functional allele, which is not so in males with only one X chromosome. IMQ does not stimulate the innate immune system of people who present these unfunctional TLR7 variants; therefore, young individuals with TLR7 mutants develop severe COVID-19, suggesting that TLR7 and TLR8 are important in SARS-CoV-2 protection [116,117]. Clearly, the presence of non-functional mutant TLR7/8 variants or reductions in their expression leads to deficient IFN-*λ* or IFN-I production [118] and is associated with severe COVID-19 [118].

In addition to these genetic susceptibility variants, people with metabolic syndrome have a deregulated immune system, with chronic low-grade inflammation. There is low TLR activation and low or null IFN induction, with higher proinflammatory ILs production favoring viral replication and the development of a cytokine storm; furthermore, systemic infection causes multi-organic damage [119,120]. We think that previous stimulation with IMQ will elevate the levels of IFNs, and further, patients will develop a protective immune response, including those with metabolic syndrome. IMQ could induce high and sustained levels of type 1 and type 3 IFNs, IL-12, and IFNg, enabling a better response if a viral infection occurs. Analyzing a great amount of coronavirus, Yang et al. report that genomic RNAs from highly pathogenic coronaviruses (MERS-CoV, SARS-CoV, and SARS-CoV-2) show weak human TLR7/8-stimulating potential [121].

## 5. Immune System Stimulation with Imiquimod

During the SARS-CoV-2 pandemic, some authors suggested the potential usefulness of IMQ as an immunostimulant to enhance individuals’ immune readiness before exposure to the virus [80,122,123,124,125,126,127]. However, there are no reported clinical studies or suggested application protocols for this specific purpose so far. Imiquimod (1-isobutyl-1H-imidazo [4,5-c]-quinolin-4-amine(C14H16N4)) is a synthetic imidazoquinoline with a molecular weight of 240.31 g·mol^−1^. IMQ acts as a TLR7/8 agonist, triggering the release of type 1 and 3 IFNs. It activates innate and adaptive immunity, offering potential applications in therapy and prophylaxis for SARS-CoV-2 infections [80], suggesting its potential role in preventing the development of LC. Initially, IMQ induces rapid innate immune activation and subsequently promotes the development of a protective Th1-type adaptive immune profile [123,124]. 

Based on existing information from its application in treating other viral diseases, IMQ may enhance the immune system, leading to improved immune responses during infections or vaccination. IMQ could serve as an adjunct treatment to expedite coronavirus clearance in the initial days of infection, reducing the time for which individuals remain contagious and avoiding the infection of internal organs.

The antibody levels against SARS-CoV-2 in vaccinated and recovered individuals decrease over time [128,129]. Moreover, the antibodies generated by vaccines exhibit a reduced neutralization capacity against new SARS-CoV-2 strains, potentially leaving individuals with immune deregulation susceptible to infection despite prior immunization [128,130,131,132,133,134]. IMQ application could enhance interferon levels, regulate anti-inflammatory and proinflammatory cytokine production, and improve antibody and cytotoxic T-lymphocyte responses, promoting a Th1 profile following post-vaccination infection or reinfection [80,124,125,127] (Figure 1).

The beneficial effects of IMQ in stimulating the immune system could induce the recovery of immunological efficiency and its self-regulation capacity, both in innate and adaptive immunity, improving both systemic and mucosal immunity; in this way, SARS-CoV-2 infection could be efficiently controlled in the upper respiratory tract in the majority of immunostimulated people, preventing the spread of the virus to the lung tissue and systemic infections; in this way, severe cases would not occur, nor organ damage, and therefore there would be no post-COVID-19 sequelae.

On the clinical front, robust and well-designed trials are essential to evaluate the immunomodulator’s safety, efficacy, and optimal dosing regimens. These trials should encompass diverse populations, accounting for variations in age, gender, comorbidities, and immune profiles. Long-term follow-up assessments are vital to ascertain the durability of the immune responses induced by the immunomodulator. A multidisciplinary approach, integrating basic science with clinical research, is imperative to unlock the full therapeutic potential of immunomodulators like IMQ in the context of SARS-CoV-2.

## 6. Effects of Imiquimod on the Innate and Adaptive Immune Systems

IMQ activates both the innate and adaptive immune systems through distinct mechanisms [135,136] (Figure 2 and Figure 3). The activation of TLR7 by viral RNA occurs in two steps and requires the presence of guanosine for the dimerization of TLR7 and the activation of the signaling cascades necessary for immunostimulation, while IMQ and other similar small molecules induce this activation on their own [110]. Due to its hydrophobic nature, non-polar vehicles are employed for its application.

Type I interferons induced by IMQ hinder viral replication, enhance NK cell activity [137,138], and stimulate monocytes–macrophages and DCs to generate nitric oxide (NO) and various cytokines, thereby boosting their phagocytic and antigen-processing capabilities [139,140,141]. IMQ elevates IFN-γ production, facilitating the development of a Th1-type immune response, while inhibiting Th2-type interleukin production [141]. Furthermore, IMQ activates B lymphocytes, promoting their differentiation and proliferation [141,142].

Like the activation of TLR7 and TLR8 by viral RNA, IMQ triggers the production of high levels of interferon-alpha (IFN-α), tumor necrosis factor-alpha (TNF-α), and various interleukins (IL-6, IL-8, IL-12) and chemokines, such as CCL2, CCL3, and CCL4 (Figure 1) [143,144,145,146,147]. When applied topically to the skin, IMQ appears to modify the immune response by activating Langerhans cells, promoting their migration, and enhancing the production of cytokines that stimulate antigen processing and presentation [147]. Beyond its local effects, topical IMQ also reinforces the immune system systemically [122]. Its application increases blood levels of type 1 interferons and other cytokines [146,147]. IMQ has the added benefit of promoting the production of IL-10, which dampens inflammation, prevents the development of cytokine storms, and supports the generation of a protective humoral immune response [145,148,149,150,151]. Interferons, IL-10 and IL-12 have anti-angiogenic properties and can induce tumor cell apoptosis [152]. 

The activation of TLR7 and TLR8 leads to the production of cytokines such as IL-1, IL-12, IL-18, IL-6, IL-10, and IFN-α, which in turn promote the generation of IFN-γ by naive T lymphocytes, favoring a Th1-type immune response and inhibiting Th2 cells (Figure 3) [142,146,153,154,155,156,157]. IMQ binds to TLR7 and TLR8 on antigen-presenting cells, activating the nuclear factor kappa-B (NF-kB) pathway through the myeloid differentiation factor 88 (MyD88)-dependent pathway; this promotes APC maturation and increases proinflammatory cytokine levels [153,157]. IMQ also induces the production of perforins in cytotoxic T cells and activates NK cells through the induction of 2′,5′-oligoadenylate synthetase.

When topically applied to treat skin cancer, IMQ activates at least 3000 genes, including those that play roles in inflammation, immune response, and the upregulation of apoptotic signaling pathways [158,159]. Consequently, through the activation of TLRs, IMQ holds significant promise in the treatment of viral infectious diseases that rely on cellular immunity. Clinical protocols for the treatment of genital warts caused by the human papillomavirus (HPV) have provided valuable insights into its beneficial and adverse effects [157,160,161,162,163,164]. Interestingly, the application of IMQ to one mucosal site of the body can induce systemic immunostimulation, including the stimulation of mucosal immunity at other sites as well [138]. Additionally, IMQ interacts as an antagonist with adenosine receptors A2AR (Figure 3), potentially contributing to its inflammatory and immunostimulatory effects [149,156]. Because immune cells excrete exosomes according to their molecular profile [165], modifying the immune profile with IMQ could exchange the miRNA profile in susceptible SARS-CoV-2 subjects, generating an efficient Th1 profile of miRNAs.

Although IMQ can be administered orally, it is most commonly available in commercial presentations as 5% and 3.5% cream for topical use (brands include Quimara, Aldara, and Mandikoz, among others). These creams contain 12.5 mg of IMQ in 250 mg of cream per packet. Ongoing research is focused on developing new application methods and vehicles to enhance its effectiveness in treating various diseases [138,166], as well as exploring other TLR7 agonists similar to IMQ, which are not yet commercially available [166,167,168,169,170,171,172].

## 7. Imiquimod as an Immunomodifier in Different Diseases

### 7.1. Experience of the Beneficial Effects of Imiquimod in Different Diseases

IMQ is approved by both the Food and Drug Administration (FDA) and the European Medicines Agency (EMA) [173,174] for the treatment of external genital warts, non-hypertrophic actinic keratoses in immunocompetent individuals, and superficial basal cell cancer (SBC) [150,175,176,177]. Beyond its approved uses, IMQ has shown promise as a therapeutic adjuvant in various types of viral-origin cancers, other neoplasms, inflammatory skin conditions, benign proliferative processes (due to its anti-angiogenic properties), and in the treatment of parasitic and chronic fungal infections, among others [178,179,180,181,182,183,184]. Although these diseases are mainly tissue-localized, several studies have shown that the immunestimulation by IMQ is systemic [122]; therefore, it may be beneficial for stimulating the respiratory tract.

In the context of eradicating HPV, clinical trials have demonstrated varying effectiveness, ranging from 30% to 75%, with a higher effectiveness observed in women [185]. Treatment with IMQ has been associated with a significant reduction in HPV virus mRNA and DNA levels [185,186]. Stimulation of the immune system, with IMQ, could facilitate the control and effective elimination of the SARS-CoV-2 virus, probably even in individuals with immunological disorders. IMQ first stimulates the innate immune system, increasing IFNs levels and further promoting the adaptive immune response to a protective profile; this stimulated immune state could help to control SARS-CoV-2 infection effectively in the upper respiratory tract, preventing it from becoming a systemic infection and thus preventing damage to internal organs such as the lungs, liver, kidneys, heart, vascular system, intestines, brain, and others, so that by not causing damage to organs, the symptoms of LC will not occur [187,188].

### 7.2. Side Effects of Imiquimod

The primary side effects associated with IMQ use in some people include topical inflammation, irritation, allergies, and notably the potential for autoimmune reactions, particularly when applied at a high dosage, frequently, or over an extended time to treat HPV warts and neoplasms [184,189,190,191]. In some cases, IMQ has been known to trigger autoimmune responses, such as psoriasis, or to exacerbate pre-existing conditions, as observed in both human and animal studies [191,192,193]. At elevated doses and when applied frequently over an extended period, IMQ has been employed to induce experimental psoriasis in both animal and human studies [190,191,194,195,196,197]. Although IMQ for the treatment of VPH warts is used at a lower dosage than that used in experimental psoriasis, it is important to exercise caution in individuals with autoimmune conditions and those predisposed to autoimmune responses. There have been isolated reports of individuals who have undergone prolonged IMQ treatments for various medical conditions and subsequently experienced severe adverse reactions following the administration of anti-SARS-CoV-2 mRNA vaccines [198].

Although SARS-CoV-2 has mechanisms to evade the action of IFNs, it appears that this viral evasion is less effective in individuals with a robust innate immune system, such as children and young adults, who largely do not develop moderate or severe cases of COVID-19 [124]. In contrast, this viral evasion seems to be more effective in individuals with a deregulated immune system. Therefore, IMQ could improve or restore the function of the immune system in people with immune deregulation, generating a better protective response against infection or through the application of vaccines. With an increase in the number of individuals with an efficient immune response, the dispersion rate of the virus in populations will be decreased [199,200].

### 7.3. Application of Imiquimod with Antiviral Vaccines

Some studies have investigated the effects of IMQ on the response to vaccination against influenza and SARS-CoV-1. Li et al. reported that combining IMQ with an influenza vaccine in mice enhances the antibody response to influenza viruses. This combination leads to increased activation and proliferation of B lymphocytes, resulting in the production of virus-specific IgM and IgG antibodies. Additionally, it elevates the expression of CD86 in mesenteric lymph nodes and spleen B lymphocytes 18 h after administration. Furthermore, this combination boosts the levels of neutralizing antibodies within three days. These findings suggest that IMQ expedites the production of antibodies induced by the influenza vaccine by promoting the rapid differentiation of naive B cells into antigen-specific antibody-producing B cells [201]. Moreover, it favors a Th1 immune profile, which enhances viral replication control and virus elimination, especially when administered intranasally [202,203]. When IMQ is applied simultaneously with the vaccine, it locally increases the production of type 1 IFNs, with systemic elevation observed from the following day onward [203]. Recent reports have shown no significant differences in the application of protein vaccines in individuals with or without IMQ as an adjuvant when applied on the same day of vaccination [204]. This apparent contradiction may be attributed to the fact that a single dose may not suffice to observe its immunostimulant effects. IMQ induces the production of type 1 IFNs early on after application, whereas the production of cytokines favoring the Th1 profile occurs later; this sequential response is essential for the development of an effective, specific immune response. IMQ is a modifier of the immune system and does not work well as an adjuvant in only one application.

There are some studies on SARS-CoV-2, including an in vitro model simulating asthma and SARS-CoV-2 infection, in which IMQ exhibited a dual effect, offering both antiviral protection by reducing ACE2 protein expression and enhancing IFN-β production and bronchial anti-inflammatory properties by decreasing proinflammatory cytokines such as IL-1β, IL-6, IL-8, and IL-33 [125]. Additionally, a study by Shen et al. compared the in vitro effects of IMQ on peripheral blood mononuclear cell (PBMC) cultures from young individuals and older adults who had received the ChAdOx1 nCoV-19 (AZD1222) vaccine. The results emphasize the importance of appropriately activating the innate immune system for the development of protective immunity in response to the ChAdOx1 nCoV-19 vaccine. These findings underscore the potential benefits of immune system stimulation in older adults [128].

Peripheral blood mononuclear cells from severe COVID-19 patients who are deficient in TLR7 genes do not respond to the in vitro application of IMQ, contrary to cell cultures from individuals with normal TLR7 genes [42]. Additionally, recent experimental models have demonstrated the positive immunostimulatory effects of IMQ when covalently bound to the S1 antigen of SARS-CoV-2 or when bound to less hydrophobic lipids. These conjugated forms of IMQ have been used as adjuvants in SARS-CoV-2 vaccines [204,205]. They have been found to enhance both the humoral and cellular immune responses against the original virus and its variants (including B.1.1.7/Alpha, B.1.351/Beta, P.1/Gamma, B.1.617.2/Delta, and B.1.1.529/Omicron), which are considered variants of concern (VOCs). Furthermore, these modified IMQ forms promote and sustain a balanced Th1/Th2 immune response.

## 8. Suggested Imiquimod Application Scheme

### 8.1. Imiquimod as an Immunostimulant

In the pursuit of optimizing immune function through the application of IMQ andconsidering that it is a treatment able to activate innate immunity first and further the adaptive immunity, an effective application scheme could entail a regimen akin to or minor to the treatment conventionally employed against HPV; this should require a dosage equivalent to the IMQ content in a 12.5 mg packet available from various commercial brands. These applications should pertain to areas of the body not exposed to direct sunlight, as recommended by previous studies [206,207,208]. Following this proposed regimen, individuals should apply one packet every three days for a total duration of four weeks, closely adhering to the manufacturer’s instructions (Table 1). Notably, the topical form of IMQ lends itself to convenient self-administration, as observed in previous studies involving willing participants [173,174].

For optimal IMQ cream application, the abdominal area stands out as a suitable choice. This selection is strategic because it facilitates the absorption of IMQ into the intestinal tissue, a pivotal location hosting a significant portion of the peripheral immune system components, notably including Peyer’s patches [209,210]. By applying IMQ to this region, the potential to stimulate both mucosal and systemic immunity is maximized. In addition to the abdominal area, other viable application sites, such as the neck and the chest, are to be considered. These regions are in proximity to the lymph nodes, which play a critical role in the early stages of SARS-CoV-2 entry and replication. The study protocols can be diversified to encompass various groups of individuals, as disparities exist in the outcomes of SARS-CoV-2 infection across different age groups, genders, and individuals with or without comorbidities.

Furthermore, the immunological profile, including the levels of IFNs type 1 and 3, along with characteristic interleukins (ILs) associated with Th1, Th2, and Th17 profiles, should be examined before, during, and after the application of IMQ. It is essential to assess the duration of the Th1 profile of specific immunity following treatment, as well as monitoring antibody levels, their ability to neutralize different viral strains, the cellular immune response, and the duration of these responses at the optimal levels. Additionally, various other fundamental aspects of the immune system should be explored to gain a comprehensive understanding of IMQ’s impact on enhancing immune responses to SARS-CoV-2.

### 8.2. Possible Side Effects of Imiquimod with the Suggested Treatment Regimen

The application of IMQ may lead to undesirable side effects in some individuals when used for the treatment of HPV infections and other conditions. These reported side effects often result from the constant and long-lasting application of IMQ, which typically extends over several weeks [211,212,213,214]. However, it is worth noting that these side effects are usually reversible once the treatment is discontinued. In the proposed scheme presented here, the frequency and duration of IMQ applications would be reduced, potentially resulting in milder or even absent side effects. It is essential to consider that in experimental models used for inducing psoriasis, both in human and animal studies, the treatment involves more frequent applications and higher doses [138,193,194,195,196,211]. Therefore, this immunostimulant scheme is less likely to predispose individuals to developing conditions like psoriasis or increase the risk of other autoimmune diseases. In the event of side effects, they are expected to be mild and temporary.

### 8.3. Contraindications

Certain individuals should be excluded from these immunostimulant treatment trial studies, including those with autoimmune diseases and individuals undergoing immunosuppressive therapies, such as transplant recipients [215,216]. An alternative treatment approach, rather than immunostimulants, should be considered for these individuals. Additionally, during treatment and the course of protocol development, individuals who experience unexpected and severe side effects, such as allergies following the initial applications, should also be excluded from this treatment regimen.

## 9. Conclusions

Stimulating the immune systems of individuals with immunological deregulation using IMQ holds the potential to significantly enhance their immune responses when confronted with SARS-CoV-2; this applies not only to those encountering the virus for the first time but also to individuals facing reinfections. IMQ may serve as a valuable immunostimulant for vaccination efforts, augmenting the protective responses generated by vaccines and supporting vaccine booster strategies.

In the context of viral infections, IMQ’s immunomodulatory effects are multifaceted. It can elevate the levels of type 1 IFNs, which play a pivotal role in inhibiting viral replication. Additionally, IMQ promotes the development of a balanced proinflammatory cytokine profile, fostering a robust innate and adaptive immune response that may result in increased concentrations of specific antibodies against SARS-CoV-2 and in enhancement of their neutralizing capacity. IMQ has the potential to boost the cytotoxic activity of CD8+ T lymphocytes and natural killer (NK) cells, bolstering the body’s antiviral defenses. IMQ could extend the duration of anti-SARS-CoV-2 antibodies and immune cells, contributing to prolonged memory immunity. Collectively, these immunostimulatory effects could lead to several desirable outcomes. They might shorten the duration of primary SARS-CoV-2 infections, reduce viral loads, expedite virus clearance during reinfections, avoid the development of systemic viral propagation, avoid organ damage, and offer improved control over infections. These immunostimulant effects would reduce the chances of those infected developing severe COVID-19 and LC. Robust and well-designed trials are indispensable to evaluating this immunomodulator’s safety, efficacy, and optimal dosing regimens in different subpopulations in the short and long term.

## Figures and Tables

**Figure 1 ijms-25-07661-f001:**
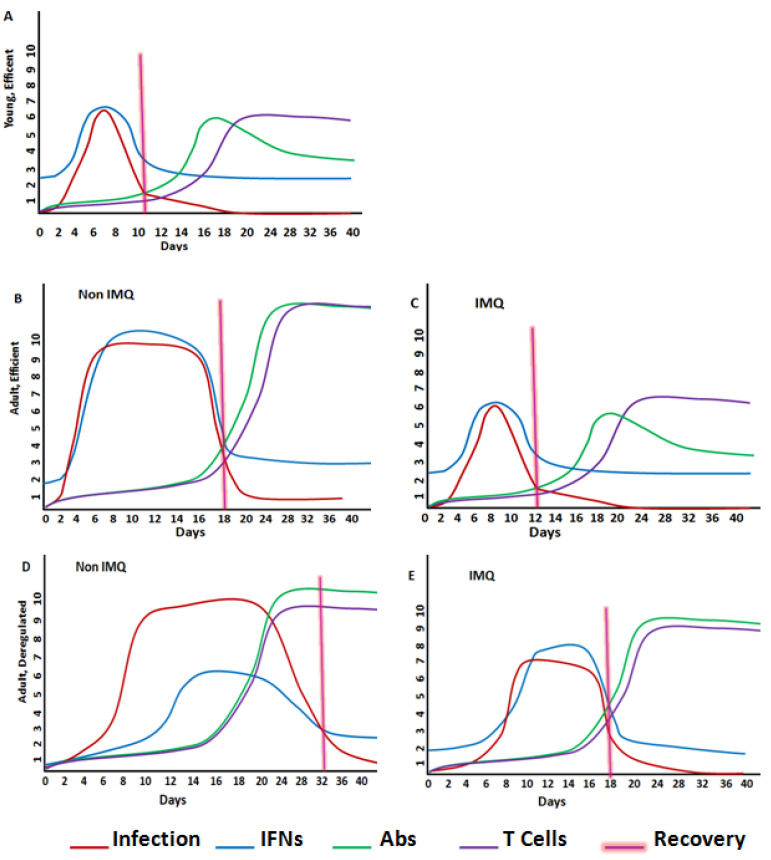
Protective immunity and deregulated immunity. (**A**) Mild SARS-CoV-2 infection is associated with early production of type I interferons (IFNs), efficient adaptive immunity, and faster resolution of viral infection. (**B**) Adult people with an efficient immune system develop moderate SARS-CoV-2 infection, associated with normal production of type I IFNs, normal humoral and cellular immunity, and increased or systemic SARS-CoV-2 proliferation, associated with highest symptoms but very low mortality [48]. (**C**) IMQ immune stimulation in adults with efficient immune system could induce elevated levels of type I IFNs, decreasing SARS-CoV-2 proliferation, and improves the humoral and cellular immune responses. (**D**) Deregulated immune system in adults develops severe SARS-CoV-2 infection, associated with delayed production of type I IFNs, slow humoral and cellular immunity, and increased systemic SARS-CoV-2 proliferation associated with more severe symptoms, with increased mortality [35]. (**E**) In those with a deregulated immune system, IMQ could restore the immune efficiency, which could result in faster resolution of COVID-19, less systemic damage, higher possibilities of survival, and lower possibilities of developing Long-COVID.

**Figure 2 ijms-25-07661-f002:**
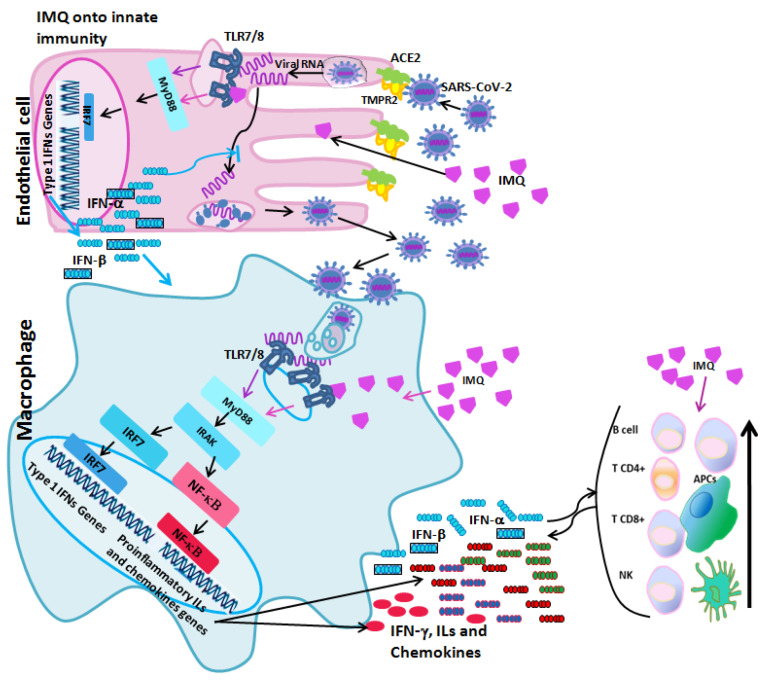
SARS-CoV-2 enters and replicates in to the ACE2+ cells. Viral RNA is recognized by TLR7/8, inducing type I and type III IFNs production that blocks viral replication. In ACE2+ cells and in innate immune cells, IMQ activates TLR7/8, increasing the production of type I and type III IFNs. High levels of type I INFs favour the blockage of viral replication and induces the production of proinflammatory interleukins, IFN-γ and chemokines that act in autocrine and paracrine manner.

**Figure 3 ijms-25-07661-f003:**
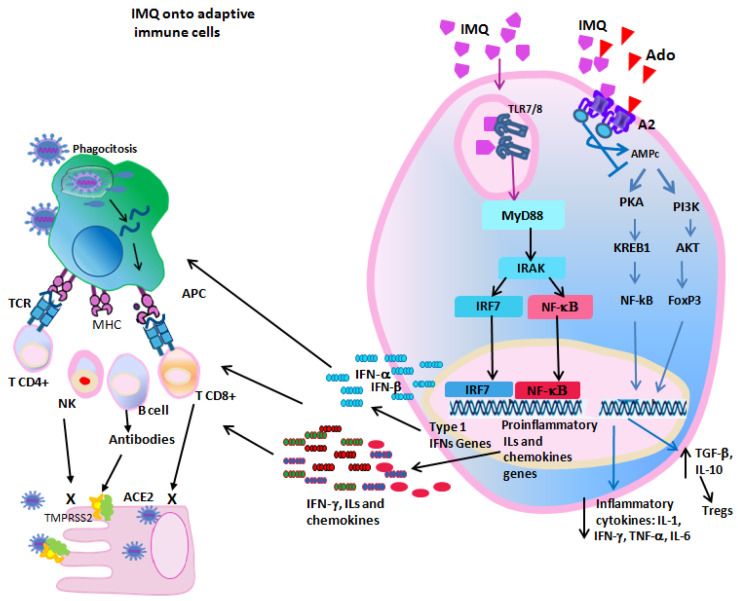
In adaptive immune cells, IMQ activates TLR7/8, increasing the production of IFNs, proinflammatory interleukins and chemokines. These act in autocrine and paracrine manner, stimulating several aspects of adaptive immune responses. IMQ also acts as an antagonist of the A2 receptor, thus blocking the adenosine-induced immunosuppression. Ado: Adenosine, A2: Adenosine receptor.

**Table 1 ijms-25-07661-t001:** Suggested imiquimod application scheme. Optimizing immune function in people with deregulated immunity through the application of IMQ; this scheme is a regimen akin to half of the treatment conventionally employed against HPV.

This entails a dosage equivalent to the IMQ content found in a 12.5 mg packet Apply the contents of one sachet (12.5 mg) to the abdominal area, adhering to the manufacturer’s instructions. Apply one packet every 3 days for a total duration of four weeks. During the treatment be aware of the degree of possible side effects. This treatment could be repeated in the days prior to the application of the vaccine, and in the first days of presenting symptoms of infection.

## Data Availability

Data sharing does not apply to this article.

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
