# Peer review of "Immune Stimulation with Imiquimod to Best Face SARS-CoV-2 Infection and Prevent Long COVID"

_ijms, 2024, doi:10.3390/ijms25147661_

Round 1

Reviewer 1 Report

Comments and Suggestions for Authors

The review “Immune stimulation with imiquimod to best face SARS-CoV-2 infection and prevent long COVID” by Ursino Pacheco-Gracia and colleagues develop the hypothesis of a possible therapy to prevent, or reduce, long COVID, a sequel of COVID. The proposed strategy is to associate imiquimod (IMQ), to enhance the immune response, to SARS-Cov2 vaccination in individuals that are potentially low responders to vaccination. Moreover, authors claim for a beneficial use of IMQ during COVID as means of reducing the risk of long COVID.

The manuscript often lacks a connection between the paragraphs (for example line 68-71), and seldom is repetitive. The manuscript requires considerable revision to be accepted for publication.

Points to be addressed:

1) In line 63 authors partially describe changes occurring in the immune function with age, why to emphasise only T cells?

2) In line 112 authors attribute a higher risk to develop long COVID to “the presence of FoxP4 locus” in certain subjects, this requires a bit more of explanation because maybe is not the locus but the expression of a certain allele/gene variants in that locus.

3) In line 120 the authors start to speak of organ damage, they refer the importance of targeting the thymus but then they move to infant COVID infections and is not clear which is the connection. And if it is the thymic dysfunction how if may affect the response to other infections/vaccinations.

4) In line 162 the authors start a chapter in which they revise the literature on the benefits/ damage of the use IMQ. I would suggest the authors to start to describe what is this molecule, line 212, also to avoid the repetition shown in the beginning of chapter 5.

5) Figure 1 is not of high quality thus numbers on the axis are not visible and colour codes are very similar leading to a bit of confusion.

6) In line 251 consider to add the abbreviation for human papilloma virus because later on is used in the text.

7) In line 253 authors state that IMQ activates 3000 genes, repeating what was previously described for cytokine secretion, etc, etc. few lines before; maybe is worth to better integrate this part of the text

8) In line 285 authors describe the use of IMQ in disease conditions in which it has been approved by FDA and EMA. Of note that these infections are quite localized and authors do not discuss how IMQ can really be used in patients with COVID infection.

9) In line 311 please make the spelling of VPH.

10) In line 326 authors refer to a study in which IMQ has been used to potentiate response to vaccination. Maybe is useful to give a reference for the work done on COVID, view that authors devote the following lines to describe response to anti-influenza vaccination in mouse models.

Considering the important role of TLR7 signal in potentiating immune response authors do not discuss much the fact, that, in the human population, there are variants of TLR7 that could jeopardize the efficiency of IMQ therapy/adjuvant. In addition, TLR8 that plays a role in the regulation of TLR7 has also been ignored in the discussion.

Author Response

1. Summary

Response and Revisions: We give corresponding response in the point-by-point response letter. The same as below:

General Comments: Comments and Suggestions for Authors:

The review “Immune stimulation with imiquimod to best face SARS-CoV-2 infection and prevent long COVID” by Ursino Pacheco-Gracia and colleagues develop the hypothesis of a possible therapy to prevent, or reduce, long COVID, a sequel of COVID. The proposed strategy is to associate imiquimod (IMQ), to enhance the immune response, to SARS-Cov2 vaccination in individuals that are potentially low responders to vaccination. Moreover, authors claim for a beneficial use of IMQ during COVID as means of reducing the risk of long COVID.

The manuscript often lacks a connection between the paragraphs (for example line 68-71), and seldom is repetitive. The manuscript requires considerable revision to be accepted for publication.

Response:

Thank you for pointing this out. We agree with this comment.

In the manuscript, we suggest that IMQ could increase the efficiency of both innate and adaptive immunity to better confront potential SARS-CoV-2 infections and reinfections, mainly in individuals with dysregulated immunity due to being of adult age or for suffering from risk factors such as metabolic syndrome (obesity, hypertension, diabetes, senescence, among others), as well as to develop better immunity induced by the application of vaccines against this coronavirus. Individuals with a stimulated immune system could control the infection in less time and prevent the development of systemic infections and organ damage, thus preventing LC.

This also makes it clear that in individuals with deficiencies in TLR7 and TLR8, this treatment would not induce this immunostimulation.

Points to be addressed:

Comments 1: In line 63 authors partially describe changes occurring in the immune function with age, why to emphasise only T cells?

Response 1: Thank you for pointing this out. We agree with this comment. Therefore, in the corrected manuscript, senescence is described more fully in this same paragraph, in lines 59 to 66.

Comments 2:  In line 112 authors attribute a higher risk to develop long COVID to “the presence of FoxP4 locus” in certain subjects, this requires a bit more of explanation because maybe is not the locus but the expression of a certain allele/gene variants in that locus.

Response 2: Thank you for pointing this out. We agree with this comment. Therefore, The information is expanded at this point, in lines 106 to 115.

Comments 3: In line 120 the authors start to speak of organ damage, they refer the importance of targeting the thymus but then they move to infant COVID infections and is not clear which is the connection. And if it is the thymic dysfunction how if may affect the response to other infections/vaccinations.

Response 3: Thank you for pointing this out. We agree with this comment. Therefore, information about the topic is added and separated from the following text, in lines 127 to 129, page 4.

Comments 4: In line 162 the authors start a chapter in which they revise the literature on the benefits/ damage of the use IMQ. I would suggest the authors to start to describe what is this molecule, line 212, also to avoid the repetition shown in the beginning of chapter 5.

Response 4: Thank you for pointing this out. We agree with this comment. Therefore, the information that was in the following chapter is adapted and added here. Chapter 5, lines 256-261.

Comments 5: Figure 1 is not of high quality thus numbers on the axis are not visible and colour codes are very similar leading to a bit of confusion.

Response 5: Thank you for pointing this out. We agree with this comment. Therefore, This figure is modified to have a better visualization.  Page 8.

Comments 6: In line 251 consider to add the abbreviation for human papilloma virus because later on is used in the text.

Response 6: Thank you for pointing this out. We agree with this comment. Therefore, This change has been made, Page 9, line 349.

Comments 7: In line 253 authors state that IMQ activates 3000 genes, repeating what was previously described for cytokine secretion, etc, etc. few lines before; maybe is worth to better integrate this part of the text

Response 7: Thank you for pointing this out. We agree with this comment. Therefore, The text is adapted as suggested.lines 344-346, Page 9.

Comments 8: In line 285 authors describe the use of IMQ in disease conditions in which it has been approved by FDA and EMA. Of note that these infections are quite localized and authors do not discuss how IMQ can really be used in patients with COVID

Response 8: Thank you for pointing this out. We agree with this comment. It is indicated that topical application of this medication has systemic effects and stimulates general immunity including mucosal immunity. Lines 385-387, page 11.

Comments 9: In line 311 please make the spelling of VPH.

Response 9: Thank you for pointing this out. We agree with this comment. Therefore, This suggestion has been made.line 408, page 12.

Comments 10: In line 326 authors refer to a study in which IMQ has been used to potentiate response to vaccination. Maybe is useful to give a reference for the work done on COVID, view that authors devote the following lines to describe response to anti-influenza vaccination in mouse models.

Response 10: Thank you for pointing this out. We agree with this comment. In lines 444-463, pge 13, onwards, two reported works on vaccines against SARS-CoV-2 are discussed in which they use IMQ linked to the S1 protein of SARS-CoV-2 and another in which they use IMQ linked to a lipid to facilitate its entry into cell membranes

Comments 11: Considering the important role of TLR7 signal in potentiating immune response authors do not discuss much the fact, that, in the human population, there are variants of TLR7 that could jeopardize the efficiency of IMQ therapy/adjuvant. In addition, TLR8 that plays a role in the regulation of TLR7 has also been ignored in the discussion. 

Response 11: Thank you for pointing this out. We agree with this comment. Therefore, A chapter is added briefly describing these TLRs. Chapter 4, pages 5 and 6.

5. Additional clarifications

1. Summary

Response and Revisions: We give corresponding response in the point-by-point response letter. The same as below.

General Comments: Comments and Suggestions for Authors:

The review “Immune stimulation with imiquimod to best face SARS-CoV-2 infection and prevent long COVID” by Ursino Pacheco-Gracia and colleagues develop the hypothesis of a possible therapy to prevent, or reduce, long COVID, a sequel of COVID. The proposed strategy is to associate imiquimod (IMQ), to enhance the immune response, to SARS-Cov2 vaccination in individuals that are potentially low responders to vaccination. Moreover, authors claim for a beneficial use of IMQ during COVID as means of reducing the risk of long COVID.

The manuscript often lacks a connection between the paragraphs (for example line 68-71), and seldom is repetitive. The manuscript requires considerable revision to be accepted for publication.

Response:

Thank you for pointing this out. We agree with this comment.

In the manuscript, we suggest that IMQ could increase the efficiency of both innate and adaptive immunity to better confront potential SARS-CoV-2 infections and reinfections, mainly in individuals with dysregulated immunity due to being of adult age or for suffering from risk factors such as metabolic syndrome (obesity, hypertension, diabetes, senescence, among others), as well as to develop better immunity induced by the application of vaccines against this coronavirus. Individuals with a stimulated immune system could control the infection in less time and prevent the development of systemic infections and organ damage, thus preventing LC.

This also makes it clear that in individuals with deficiencies in TLR7 and TLR8, this treatment would not induce this immunostimulation.

Points to be addressed:

Comments 1: In line 63 authors partially describe changes occurring in the immune function with age, why to emphasise only T cells?

Response 1: Thank you for pointing this out. We agree with this comment. Therefore, in the corrected manuscript, senescence is described more fully in this same paragraph, in lines 59 to 66.

Comments 2:  In line 112 authors attribute a higher risk to develop long COVID to “the presence of FoxP4 locus” in certain subjects, this requires a bit more of explanation because maybe is not the locus but the expression of a certain allele/gene variants in that locus.

Response 2: Thank you for pointing this out. We agree with this comment. Therefore, The information is expanded at this point, in lines 106 to 115.

Comments 3: In line 120 the authors start to speak of organ damage, they refer the importance of targeting the thymus but then they move to infant COVID infections and is not clear which is the connection. And if it is the thymic dysfunction how if may affect the response to other infections/vaccinations.

Response 3: Thank you for pointing this out. We agree with this comment. Therefore, information about the topic is added and separated from the following text, in lines 127 to 129, page 4.

Comments 4: In line 162 the authors start a chapter in which they revise the literature on the benefits/ damage of the use IMQ. I would suggest the authors to start to describe what is this molecule, line 212, also to avoid the repetition shown in the beginning of chapter 5.

Response 4: Thank you for pointing this out. We agree with this comment. Therefore, the information that was in the following chapter is adapted and added here. Chapter 5, lines 256-261.

Comments 5: Figure 1 is not of high quality thus numbers on the axis are not visible and colour codes are very similar leading to a bit of confusion.

Response 5: Thank you for pointing this out. We agree with this comment. Therefore, This figure is modified to have a better visualization.  Page 8.

Comments 6: In line 251 consider to add the abbreviation for human papilloma virus because later on is used in the text.

Response 6: Thank you for pointing this out. We agree with this comment. Therefore, This change has been made, Page 9, line 349.

Comments 7: In line 253 authors state that IMQ activates 3000 genes, repeating what was previously described for cytokine secretion, etc, etc. few lines before; maybe is worth to better integrate this part of the text

Response 7: Thank you for pointing this out. We agree with this comment. Therefore, The text is adapted as suggested.lines 344-346, Page 9.

Comments 8: In line 285 authors describe the use of IMQ in disease conditions in which it has been approved by FDA and EMA. Of note that these infections are quite localized and authors do not discuss how IMQ can really be used in patients with COVID

Response 8: Thank you for pointing this out. We agree with this comment. It is indicated that topical application of this medication has systemic effects and stimulates general immunity including mucosal immunity. Lines 385-387, page 11.

Comments 9: In line 311 please make the spelling of VPH.

Response 9: Thank you for pointing this out. We agree with this comment. Therefore, This suggestion has been made.line 408, page 12.

Comments 10: In line 326 authors refer to a study in which IMQ has been used to potentiate response to vaccination. Maybe is useful to give a reference for the work done on COVID, view that authors devote the following lines to describe response to anti-influenza vaccination in mouse models.

Response 10: Thank you for pointing this out. We agree with this comment. In lines 444-463, pge 13, onwards, two reported works on vaccines against SARS-CoV-2 are discussed in which they use IMQ linked to the S1 protein of SARS-CoV-2 and another in which they use IMQ linked to a lipid to facilitate its entry into cell membranes

Comments 11: Considering the important role of TLR7 signal in potentiating immune response authors do not discuss much the fact, that, in the human population, there are variants of TLR7 that could jeopardize the efficiency of IMQ therapy/adjuvant. In addition, TLR8 that plays a role in the regulation of TLR7 has also been ignored in the discussion. 

Response 11: Thank you for pointing this out. We agree with this comment. Therefore, A chapter is added briefly describing these TLRs. Chapter 4, pages 5 and 6.

5. Additional clarifications

1. Summary

Response and Revisions: We give corresponding response in the point-by-point response letter. The same as below.

General Comments: Comments and Suggestions for Authors:

The review “Immune stimulation with imiquimod to best face SARS-CoV-2 infection and prevent long COVID” by Ursino Pacheco-Gracia and colleagues develop the hypothesis of a possible therapy to prevent, or reduce, long COVID, a sequel of COVID. The proposed strategy is to associate imiquimod (IMQ), to enhance the immune response, to SARS-Cov2 vaccination in individuals that are potentially low responders to vaccination. Moreover, authors claim for a beneficial use of IMQ during COVID as means of reducing the risk of long COVID.

The manuscript often lacks a connection between the paragraphs (for example line 68-71), and seldom is repetitive. The manuscript requires considerable revision to be accepted for publication.

Response:

Thank you for pointing this out. We agree with this comment.

In the manuscript, we suggest that IMQ could increase the efficiency of both innate and adaptive immunity to better confront potential SARS-CoV-2 infections and reinfections, mainly in individuals with dysregulated immunity due to being of adult age or for suffering from risk factors such as metabolic syndrome (obesity, hypertension, diabetes, senescence, among others), as well as to develop better immunity induced by the application of vaccines against this coronavirus. Individuals with a stimulated immune system could control the infection in less time and prevent the development of systemic infections and organ damage, thus preventing LC.

This also makes it clear that in individuals with deficiencies in TLR7 and TLR8, this treatment would not induce this immunostimulation.

Points to be addressed:

Comments 1: In line 63 authors partially describe changes occurring in the immune function with age, why to emphasise only T cells?

Response 1: Thank you for pointing this out. We agree with this comment. Therefore, in the corrected manuscript, senescence is described more fully in this same paragraph, in lines 59 to 66.

Comments 2:  In line 112 authors attribute a higher risk to develop long COVID to “the presence of FoxP4 locus” in certain subjects, this requires a bit more of explanation because maybe is not the locus but the expression of a certain allele/gene variants in that locus.

Response 2: Thank you for pointing this out. We agree with this comment. Therefore, The information is expanded at this point, in lines 106 to 115.

Comments 3: In line 120 the authors start to speak of organ damage, they refer the importance of targeting the thymus but then they move to infant COVID infections and is not clear which is the connection. And if it is the thymic dysfunction how if may affect the response to other infections/vaccinations.

Response 3: Thank you for pointing this out. We agree with this comment. Therefore, information about the topic is added and separated from the following text, in lines 127 to 129, page 4.

Comments 4: In line 162 the authors start a chapter in which they revise the literature on the benefits/ damage of the use IMQ. I would suggest the authors to start to describe what is this molecule, line 212, also to avoid the repetition shown in the beginning of chapter 5.

Response 4: Thank you for pointing this out. We agree with this comment. Therefore, the information that was in the following chapter is adapted and added here. Chapter 5, lines 256-261.

Comments 5: Figure 1 is not of high quality thus numbers on the axis are not visible and colour codes are very similar leading to a bit of confusion.

Response 5: Thank you for pointing this out. We agree with this comment. Therefore, This figure is modified to have a better visualization.  Page 8.

Comments 6: In line 251 consider to add the abbreviation for human papilloma virus because later on is used in the text.

Response 6: Thank you for pointing this out. We agree with this comment. Therefore, This change has been made, Page 9, line 349.

Comments 7: In line 253 authors state that IMQ activates 3000 genes, repeating what was previously described for cytokine secretion, etc, etc. few lines before; maybe is worth to better integrate this part of the text

Response 7: Thank you for pointing this out. We agree with this comment. Therefore, The text is adapted as suggested.lines 344-346, Page 9.

Comments 8: In line 285 authors describe the use of IMQ in disease conditions in which it has been approved by FDA and EMA. Of note that these infections are quite localized and authors do not discuss how IMQ can really be used in patients with COVID

Response 8: Thank you for pointing this out. We agree with this comment. It is indicated that topical application of this medication has systemic effects and stimulates general immunity including mucosal immunity. Lines 385-387, page 11.

Comments 9: In line 311 please make the spelling of VPH.

Response 9: Thank you for pointing this out. We agree with this comment. Therefore, This suggestion has been made.line 408, page 12.

Comments 10: In line 326 authors refer to a study in which IMQ has been used to potentiate response to vaccination. Maybe is useful to give a reference for the work done on COVID, view that authors devote the following lines to describe response to anti-influenza vaccination in mouse models.

Response 10: Thank you for pointing this out. We agree with this comment. In lines 444-463, pge 13, onwards, two reported works on vaccines against SARS-CoV-2 are discussed in which they use IMQ linked to the S1 protein of SARS-CoV-2 and another in which they use IMQ linked to a lipid to facilitate its entry into cell membranes

Comments 11: Considering the important role of TLR7 signal in potentiating immune response authors do not discuss much the fact, that, in the human population, there are variants of TLR7 that could jeopardize the efficiency of IMQ therapy/adjuvant. In addition, TLR8 that plays a role in the regulation of TLR7 has also been ignored in the discussion. 

Response 11: Thank you for pointing this out. We agree with this comment. Therefore, A chapter is added briefly describing these TLRs. Chapter 4, pages 5 and 6.

5. Additional clarifications

1. Summary

Response and Revisions: We give corresponding response in the point-by-point response letter. The same as below.

General Comments: Comments and Suggestions for Authors:

The review “Immune stimulation with imiquimod to best face SARS-CoV-2 infection and prevent long COVID” by Ursino Pacheco-Gracia and colleagues develop the hypothesis of a possible therapy to prevent, or reduce, long COVID, a sequel of COVID. The proposed strategy is to associate imiquimod (IMQ), to enhance the immune response, to SARS-Cov2 vaccination in individuals that are potentially low responders to vaccination. Moreover, authors claim for a beneficial use of IMQ during COVID as means of reducing the risk of long COVID.

The manuscript often lacks a connection between the paragraphs (for example line 68-71), and seldom is repetitive. The manuscript requires considerable revision to be accepted for publication.

Response:

Thank you for pointing this out. We agree with this comment.

In the manuscript, we suggest that IMQ could increase the efficiency of both innate and adaptive immunity to better confront potential SARS-CoV-2 infections and reinfections, mainly in individuals with dysregulated immunity due to being of adult age or for suffering from risk factors such as metabolic syndrome (obesity, hypertension, diabetes, senescence, among others), as well as to develop better immunity induced by the application of vaccines against this coronavirus. Individuals with a stimulated immune system could control the infection in less time and prevent the development of systemic infections and organ damage, thus preventing LC.

This also makes it clear that in individuals with deficiencies in TLR7 and TLR8, this treatment would not induce this immunostimulation.

Points to be addressed:

Comments 1: In line 63 authors partially describe changes occurring in the immune function with age, why to emphasise only T cells?

Response 1: Thank you for pointing this out. We agree with this comment. Therefore, in the corrected manuscript, senescence is described more fully in this same paragraph, in lines 59 to 66.

Comments 2:  In line 112 authors attribute a higher risk to develop long COVID to “the presence of FoxP4 locus” in certain subjects, this requires a bit more of explanation because maybe is not the locus but the expression of a certain allele/gene variants in that locus.

Response 2: Thank you for pointing this out. We agree with this comment. Therefore, The information is expanded at this point, in lines 106 to 115.

Comments 3: In line 120 the authors start to speak of organ damage, they refer the importance of targeting the thymus but then they move to infant COVID infections and is not clear which is the connection. And if it is the thymic dysfunction how if may affect the response to other infections/vaccinations.

Response 3: Thank you for pointing this out. We agree with this comment. Therefore, information about the topic is added and separated from the following text, in lines 127 to 129, page 4.

Comments 4: In line 162 the authors start a chapter in which they revise the literature on the benefits/ damage of the use IMQ. I would suggest the authors to start to describe what is this molecule, line 212, also to avoid the repetition shown in the beginning of chapter 5.

Response 4: Thank you for pointing this out. We agree with this comment. Therefore, the information that was in the following chapter is adapted and added here. Chapter 5, lines 256-261.

Comments 5: Figure 1 is not of high quality thus numbers on the axis are not visible and colour codes are very similar leading to a bit of confusion.

Response 5: Thank you for pointing this out. We agree with this comment. Therefore, This figure is modified to have a better visualization.  Page 8.

Comments 6: In line 251 consider to add the abbreviation for human papilloma virus because later on is used in the text.

Response 6: Thank you for pointing this out. We agree with this comment. Therefore, This change has been made, Page 9, line 349.

Comments 7: In line 253 authors state that IMQ activates 3000 genes, repeating what was previously described for cytokine secretion, etc, etc. few lines before; maybe is worth to better integrate this part of the text

Response 7: Thank you for pointing this out. We agree with this comment. Therefore, The text is adapted as suggested.lines 344-346, Page 9.

Comments 8: In line 285 authors describe the use of IMQ in disease conditions in which it has been approved by FDA and EMA. Of note that these infections are quite localized and authors do not discuss how IMQ can really be used in patients with COVID

Response 8: Thank you for pointing this out. We agree with this comment. It is indicated that topical application of this medication has systemic effects and stimulates general immunity including mucosal immunity. Lines 385-387, page 11.

Comments 9: In line 311 please make the spelling of VPH.

Response 9: Thank you for pointing this out. We agree with this comment. Therefore, This suggestion has been made.line 408, page 12.

Comments 10: In line 326 authors refer to a study in which IMQ has been used to potentiate response to vaccination. Maybe is useful to give a reference for the work done on COVID, view that authors devote the following lines to describe response to anti-influenza vaccination in mouse models.

Response 10: Thank you for pointing this out. We agree with this comment. In lines 444-463, pge 13, onwards, two reported works on vaccines against SARS-CoV-2 are discussed in which they use IMQ linked to the S1 protein of SARS-CoV-2 and another in which they use IMQ linked to a lipid to facilitate its entry into cell membranes

Comments 11: Considering the important role of TLR7 signal in potentiating immune response authors do not discuss much the fact, that, in the human population, there are variants of TLR7 that could jeopardize the efficiency of IMQ therapy/adjuvant. In addition, TLR8 that plays a role in the regulation of TLR7 has also been ignored in the discussion. 

Response 11: Thank you for pointing this out. We agree with this comment. Therefore, A chapter is added briefly describing these TLRs. Chapter 4, pages 5 and 6.

5. Additional clarifications

1. Summary

Response and Revisions: We give corresponding response in the point-by-point response letter. The same as below.

General Comments: Comments and Suggestions for Authors:

The review “Immune stimulation with imiquimod to best face SARS-CoV-2 infection and prevent long COVID” by Ursino Pacheco-Gracia and colleagues develop the hypothesis of a possible therapy to prevent, or reduce, long COVID, a sequel of COVID. The proposed strategy is to associate imiquimod (IMQ), to enhance the immune response, to SARS-Cov2 vaccination in individuals that are potentially low responders to vaccination. Moreover, authors claim for a beneficial use of IMQ during COVID as means of reducing the risk of long COVID.

The manuscript often lacks a connection between the paragraphs (for example line 68-71), and seldom is repetitive. The manuscript requires considerable revision to be accepted for publication.

Response:

Thank you for pointing this out. We agree with this comment.

In the manuscript, we suggest that IMQ could increase the efficiency of both innate and adaptive immunity to better confront potential SARS-CoV-2 infections and reinfections, mainly in individuals with dysregulated immunity due to being of adult age or for suffering from risk factors such as metabolic syndrome (obesity, hypertension, diabetes, senescence, among others), as well as to develop better immunity induced by the application of vaccines against this coronavirus. Individuals with a stimulated immune system could control the infection in less time and prevent the development of systemic infections and organ damage, thus preventing LC.

This also makes it clear that in individuals with deficiencies in TLR7 and TLR8, this treatment would not induce this immunostimulation.

Points to be addressed:

Comments 1: In line 63 authors partially describe changes occurring in the immune function with age, why to emphasise only T cells?

Response 1: Thank you for pointing this out. We agree with this comment. Therefore, in the corrected manuscript, senescence is described more fully in this same paragraph, in lines 59 to 66.

Comments 2:  In line 112 authors attribute a higher risk to develop long COVID to “the presence of FoxP4 locus” in certain subjects, this requires a bit more of explanation because maybe is not the locus but the expression of a certain allele/gene variants in that locus.

Response 2: Thank you for pointing this out. We agree with this comment. Therefore, The information is expanded at this point, in lines 106 to 115.

Comments 3: In line 120 the authors start to speak of organ damage, they refer the importance of targeting the thymus but then they move to infant COVID infections and is not clear which is the connection. And if it is the thymic dysfunction how if may affect the response to other infections/vaccinations.

Response 3: Thank you for pointing this out. We agree with this comment. Therefore, information about the topic is added and separated from the following text, in lines 127 to 129, page 4.

Comments 4: In line 162 the authors start a chapter in which they revise the literature on the benefits/ damage of the use IMQ. I would suggest the authors to start to describe what is this molecule, line 212, also to avoid the repetition shown in the beginning of chapter 5.

Response 4: Thank you for pointing this out. We agree with this comment. Therefore, the information that was in the following chapter is adapted and added here. Chapter 5, lines 256-261.

Comments 5: Figure 1 is not of high quality thus numbers on the axis are not visible and colour codes are very similar leading to a bit of confusion.

Response 5: Thank you for pointing this out. We agree with this comment. Therefore, This figure is modified to have a better visualization.  Page 8.

Comments 6: In line 251 consider to add the abbreviation for human papilloma virus because later on is used in the text.

Response 6: Thank you for pointing this out. We agree with this comment. Therefore, This change has been made, Page 9, line 349.

Comments 7: In line 253 authors state that IMQ activates 3000 genes, repeating what was previously described for cytokine secretion, etc, etc. few lines before; maybe is worth to better integrate this part of the text

Response 7: Thank you for pointing this out. We agree with this comment. Therefore, The text is adapted as suggested.lines 344-346, Page 9.

Comments 8: In line 285 authors describe the use of IMQ in disease conditions in which it has been approved by FDA and EMA. Of note that these infections are quite localized and authors do not discuss how IMQ can really be used in patients with COVID

Response 8: Thank you for pointing this out. We agree with this comment. It is indicated that topical application of this medication has systemic effects and stimulates general immunity including mucosal immunity. Lines 385-387, page 11.

Comments 9: In line 311 please make the spelling of VPH.

Response 9: Thank you for pointing this out. We agree with this comment. Therefore, This suggestion has been made.line 408, page 12.

Comments 10: In line 326 authors refer to a study in which IMQ has been used to potentiate response to vaccination. Maybe is useful to give a reference for the work done on COVID, view that authors devote the following lines to describe response to anti-influenza vaccination in mouse models.

Response 10: Thank you for pointing this out. We agree with this comment. In lines 444-463, pge 13, onwards, two reported works on vaccines against SARS-CoV-2 are discussed in which they use IMQ linked to the S1 protein of SARS-CoV-2 and another in which they use IMQ linked to a lipid to facilitate its entry into cell membranes

Comments 11: Considering the important role of TLR7 signal in potentiating immune response authors do not discuss much the fact, that, in the human population, there are variants of TLR7 that could jeopardize the efficiency of IMQ therapy/adjuvant. In addition, TLR8 that plays a role in the regulation of TLR7 has also been ignored in the discussion. 

Response 11: Thank you for pointing this out. We agree with this comment. Therefore, A chapter is added briefly describing these TLRs. Chapter 4, pages 5 and 6.

5. Additional clarifications

1. Summary

Response and Revisions: We give corresponding response in the point-by-point response letter. The same as below.

General Comments: Comments and Suggestions for Authors:

The review “Immune stimulation with imiquimod to best face SARS-CoV-2 infection and prevent long COVID” by Ursino Pacheco-Gracia and colleagues develop the hypothesis of a possible therapy to prevent, or reduce, long COVID, a sequel of COVID. The proposed strategy is to associate imiquimod (IMQ), to enhance the immune response, to SARS-Cov2 vaccination in individuals that are potentially low responders to vaccination. Moreover, authors claim for a beneficial use of IMQ during COVID as means of reducing the risk of long COVID.

The manuscript often lacks a connection between the paragraphs (for example line 68-71), and seldom is repetitive. The manuscript requires considerable revision to be accepted for publication.

Response:

Thank you for pointing this out. We agree with this comment.

In the manuscript, we suggest that IMQ could increase the efficiency of both innate and adaptive immunity to better confront potential SARS-CoV-2 infections and reinfections, mainly in individuals with dysregulated immunity due to being of adult age or for suffering from risk factors such as metabolic syndrome (obesity, hypertension, diabetes, senescence, among others), as well as to develop better immunity induced by the application of vaccines against this coronavirus. Individuals with a stimulated immune system could control the infection in less time and prevent the development of systemic infections and organ damage, thus preventing LC.

This also makes it clear that in individuals with deficiencies in TLR7 and TLR8, this treatment would not induce this immunostimulation.

Points to be addressed:

Comments 1: In line 63 authors partially describe changes occurring in the immune function with age, why to emphasise only T cells?

Response 1: Thank you for pointing this out. We agree with this comment. Therefore, in the corrected manuscript, senescence is described more fully in this same paragraph, in lines 59 to 66.

Comments 2:  In line 112 authors attribute a higher risk to develop long COVID to “the presence of FoxP4 locus” in certain subjects, this requires a bit more of explanation because maybe is not the locus but the expression of a certain allele/gene variants in that locus.

Response 2: Thank you for pointing this out. We agree with this comment. Therefore, The information is expanded at this point, in lines 106 to 115.

Comments 3: In line 120 the authors start to speak of organ damage, they refer the importance of targeting the thymus but then they move to infant COVID infections and is not clear which is the connection. And if it is the thymic dysfunction how if may affect the response to other infections/vaccinations.

Response 3: Thank you for pointing this out. We agree with this comment. Therefore, information about the topic is added and separated from the following text, in lines 127 to 129, page 4.

Comments 4: In line 162 the authors start a chapter in which they revise the literature on the benefits/ damage of the use IMQ. I would suggest the authors to start to describe what is this molecule, line 212, also to avoid the repetition shown in the beginning of chapter 5.

Response 4: Thank you for pointing this out. We agree with this comment. Therefore, the information that was in the following chapter is adapted and added here. Chapter 5, lines 256-261.

Comments 5: Figure 1 is not of high quality thus numbers on the axis are not visible and colour codes are very similar leading to a bit of confusion.

Response 5: Thank you for pointing this out. We agree with this comment. Therefore, This figure is modified to have a better visualization.  Page 8.

Comments 6: In line 251 consider to add the abbreviation for human papilloma virus because later on is used in the text.

Response 6: Thank you for pointing this out. We agree with this comment. Therefore, This change has been made, Page 9, line 349.

Comments 7: In line 253 authors state that IMQ activates 3000 genes, repeating what was previously described for cytokine secretion, etc, etc. few lines before; maybe is worth to better integrate this part of the text

Response 7: Thank you for pointing this out. We agree with this comment. Therefore, The text is adapted as suggested.lines 344-346, Page 9.

Comments 8: In line 285 authors describe the use of IMQ in disease conditions in which it has been approved by FDA and EMA. Of note that these infections are quite localized and authors do not discuss how IMQ can really be used in patients with COVID

Response 8: Thank you for pointing this out. We agree with this comment. It is indicated that topical application of this medication has systemic effects and stimulates general immunity including mucosal immunity. Lines 385-387, page 11.

Comments 9: In line 311 please make the spelling of VPH.

Response 9: Thank you for pointing this out. We agree with this comment. Therefore, This suggestion has been made.line 408, page 12.

Comments 10: In line 326 authors refer to a study in which IMQ has been used to potentiate response to vaccination. Maybe is useful to give a reference for the work done on COVID, view that authors devote the following lines to describe response to anti-influenza vaccination in mouse models.

Response 10: Thank you for pointing this out. We agree with this comment. In lines 444-463, pge 13, onwards, two reported works on vaccines against SARS-CoV-2 are discussed in which they use IMQ linked to the S1 protein of SARS-CoV-2 and another in which they use IMQ linked to a lipid to facilitate its entry into cell membranes

Comments 11: Considering the important role of TLR7 signal in potentiating immune response authors do not discuss much the fact, that, in the human population, there are variants of TLR7 that could jeopardize the efficiency of IMQ therapy/adjuvant. In addition, TLR8 that plays a role in the regulation of TLR7 has also been ignored in the discussion. 

Response 11: Thank you for pointing this out. We agree with this comment. Therefore, A chapter is added briefly describing these TLRs. Chapter 4, pages 5 and 6.

5. Additional clarifications

1. Summary

Response and Revisions: We give corresponding response in the point-by-point response letter. The same as below.

General Comments: Comments and Suggestions for Authors:

The review “Immune stimulation with imiquimod to best face SARS-CoV-2 infection and prevent long COVID” by Ursino Pacheco-Gracia and colleagues develop the hypothesis of a possible therapy to prevent, or reduce, long COVID, a sequel of COVID. The proposed strategy is to associate imiquimod (IMQ), to enhance the immune response, to SARS-Cov2 vaccination in individuals that are potentially low responders to vaccination. Moreover, authors claim for a beneficial use of IMQ during COVID as means of reducing the risk of long COVID.

The manuscript often lacks a connection between the paragraphs (for example line 68-71), and seldom is repetitive. The manuscript requires considerable revision to be accepted for publication.

Response:

Thank you for pointing this out. We agree with this comment.

In the manuscript, we suggest that IMQ could increase the efficiency of both innate and adaptive immunity to better confront potential SARS-CoV-2 infections and reinfections, mainly in individuals with dysregulated immunity due to being of adult age or for suffering from risk factors such as metabolic syndrome (obesity, hypertension, diabetes, senescence, among others), as well as to develop better immunity induced by the application of vaccines against this coronavirus. Individuals with a stimulated immune system could control the infection in less time and prevent the development of systemic infections and organ damage, thus preventing LC.

This also makes it clear that in individuals with deficiencies in TLR7 and TLR8, this treatment would not induce this immunostimulation.

Points to be addressed:

Comments 1: In line 63 authors partially describe changes occurring in the immune function with age, why to emphasise only T cells?

Response 1: Thank you for pointing this out. We agree with this comment. Therefore, in the corrected manuscript, senescence is described more fully in this same paragraph, in lines 59 to 66.

Comments 2:  In line 112 authors attribute a higher risk to develop long COVID to “the presence of FoxP4 locus” in certain subjects, this requires a bit more of explanation because maybe is not the locus but the expression of a certain allele/gene variants in that locus.

Response 2: Thank you for pointing this out. We agree with this comment. Therefore, The information is expanded at this point, in lines 106 to 115.

Comments 3: In line 120 the authors start to speak of organ damage, they refer the importance of targeting the thymus but then they move to infant COVID infections and is not clear which is the connection. And if it is the thymic dysfunction how if may affect the response to other infections/vaccinations.

Response 3: Thank you for pointing this out. We agree with this comment. Therefore, information about the topic is added and separated from the following text, in lines 127 to 129, page 4.

Comments 4: In line 162 the authors start a chapter in which they revise the literature on the benefits/ damage of the use IMQ. I would suggest the authors to start to describe what is this molecule, line 212, also to avoid the repetition shown in the beginning of chapter 5.

Response 4: Thank you for pointing this out. We agree with this comment. Therefore, the information that was in the following chapter is adapted and added here. Chapter 5, lines 256-261.

Comments 5: Figure 1 is not of high quality thus numbers on the axis are not visible and colour codes are very similar leading to a bit of confusion.

Response 5: Thank you for pointing this out. We agree with this comment. Therefore, This figure is modified to have a better visualization.  Page 8.

Comments 6: In line 251 consider to add the abbreviation for human papilloma virus because later on is used in the text.

Response 6: Thank you for pointing this out. We agree with this comment. Therefore, This change has been made, Page 9, line 349.

Comments 7: In line 253 authors state that IMQ activates 3000 genes, repeating what was previously described for cytokine secretion, etc, etc. few lines before; maybe is worth to better integrate this part of the text

Response 7: Thank you for pointing this out. We agree with this comment. Therefore, The text is adapted as suggested.lines 344-346, Page 9.

Comments 8: In line 285 authors describe the use of IMQ in disease conditions in which it has been approved by FDA and EMA. Of note that these infections are quite localized and authors do not discuss how IMQ can really be used in patients with COVID

Response 8: Thank you for pointing this out. We agree with this comment. It is indicated that topical application of this medication has systemic effects and stimulates general immunity including mucosal immunity. Lines 385-387, page 11.

Comments 9: In line 311 please make the spelling of VPH.

Response 9: Thank you for pointing this out. We agree with this comment. Therefore, This suggestion has been made.line 408, page 12.

Comments 10: In line 326 authors refer to a study in which IMQ has been used to potentiate response to vaccination. Maybe is useful to give a reference for the work done on COVID, view that authors devote the following lines to describe response to anti-influenza vaccination in mouse models.

Response 10: Thank you for pointing this out. We agree with this comment. In lines 444-463, pge 13, onwards, two reported works on vaccines against SARS-CoV-2 are discussed in which they use IMQ linked to the S1 protein of SARS-CoV-2 and another in which they use IMQ linked to a lipid to facilitate its entry into cell membranes

Comments 11: Considering the important role of TLR7 signal in potentiating immune response authors do not discuss much the fact, that, in the human population, there are variants of TLR7 that could jeopardize the efficiency of IMQ therapy/adjuvant. In addition, TLR8 that plays a role in the regulation of TLR7 has also been ignored in the discussion. 

Response 11: Thank you for pointing this out. We agree with this comment. Therefore, A chapter is added briefly describing these TLRs. Chapter 4, pages 5 and 6.

5. Additional clarifications

1. Summary

Response and Revisions: We give corresponding response in the point-by-point response letter. The same as below.

General Comments: Comments and Suggestions for Authors:

The review “Immune stimulation with imiquimod to best face SARS-CoV-2 infection and prevent long COVID” by Ursino Pacheco-Gracia and colleagues develop the hypothesis of a possible therapy to prevent, or reduce, long COVID, a sequel of COVID. The proposed strategy is to associate imiquimod (IMQ), to enhance the immune response, to SARS-Cov2 vaccination in individuals that are potentially low responders to vaccination. Moreover, authors claim for a beneficial use of IMQ during COVID as means of reducing the risk of long COVID.

The manuscript often lacks a connection between the paragraphs (for example line 68-71), and seldom is repetitive. The manuscript requires considerable revision to be accepted for publication.

Response:

Thank you for pointing this out. We agree with this comment.

In the manuscript, we suggest that IMQ could increase the efficiency of both innate and adaptive immunity to better confront potential SARS-CoV-2 infections and reinfections, mainly in individuals with dysregulated immunity due to being of adult age or for suffering from risk factors such as metabolic syndrome (obesity, hypertension, diabetes, senescence, among others), as well as to develop better immunity induced by the application of vaccines against this coronavirus. Individuals with a stimulated immune system could control the infection in less time and prevent the development of systemic infections and organ damage, thus preventing LC.

This also makes it clear that in individuals with deficiencies in TLR7 and TLR8, this treatment would not induce this immunostimulation.

Points to be addressed:

Comments 1: In line 63 authors partially describe changes occurring in the immune function with age, why to emphasise only T cells?

Response 1: Thank you for pointing this out. We agree with this comment. Therefore, in the corrected manuscript, senescence is described more fully in this same paragraph, in lines 59 to 66.

Comments 2:  In line 112 authors attribute a higher risk to develop long COVID to “the presence of FoxP4 locus” in certain subjects, this requires a bit more of explanation because maybe is not the locus but the expression of a certain allele/gene variants in that locus.

Response 2: Thank you for pointing this out. We agree with this comment. Therefore, The information is expanded at this point, in lines 106 to 115.

Comments 3: In line 120 the authors start to speak of organ damage, they refer the importance of targeting the thymus but then they move to infant COVID infections and is not clear which is the connection. And if it is the thymic dysfunction how if may affect the response to other infections/vaccinations.

Response 3: Thank you for pointing this out. We agree with this comment. Therefore, information about the topic is added and separated from the following text, in lines 127 to 129, page 4.

Comments 4: In line 162 the authors start a chapter in which they revise the literature on the benefits/ damage of the use IMQ. I would suggest the authors to start to describe what is this molecule, line 212, also to avoid the repetition shown in the beginning of chapter 5.

Response 4: Thank you for pointing this out. We agree with this comment. Therefore, the information that was in the following chapter is adapted and added here. Chapter 5, lines 256-261.

Comments 5: Figure 1 is not of high quality thus numbers on the axis are not visible and colour codes are very similar leading to a bit of confusion.

Response 5: Thank you for pointing this out. We agree with this comment. Therefore, This figure is modified to have a better visualization.  Page 8.

Comments 6: In line 251 consider to add the abbreviation for human papilloma virus because later on is used in the text.

Response 6: Thank you for pointing this out. We agree with this comment. Therefore, This change has been made, Page 9, line 349.

Comments 7: In line 253 authors state that IMQ activates 3000 genes, repeating what was previously described for cytokine secretion, etc, etc. few lines before; maybe is worth to better integrate this part of the text

Response 7: Thank you for pointing this out. We agree with this comment. Therefore, The text is adapted as suggested.lines 344-346, Page 9.

Comments 8: In line 285 authors describe the use of IMQ in disease conditions in which it has been approved by FDA and EMA. Of note that these infections are quite localized and authors do not discuss how IMQ can really be used in patients with COVID

Response 8: Thank you for pointing this out. We agree with this comment. It is indicated that topical application of this medication has systemic effects and stimulates general immunity including mucosal immunity. Lines 385-387, page 11.

Comments 9: In line 311 please make the spelling of VPH.

Response 9: Thank you for pointing this out. We agree with this comment. Therefore, This suggestion has been made.line 408, page 12.

Comments 10: In line 326 authors refer to a study in which IMQ has been used to potentiate response to vaccination. Maybe is useful to give a reference for the work done on COVID, view that authors devote the following lines to describe response to anti-influenza vaccination in mouse models.

Response 10: Thank you for pointing this out. We agree with this comment. In lines 444-463, pge 13, onwards, two reported works on vaccines against SARS-CoV-2 are discussed in which they use IMQ linked to the S1 protein of SARS-CoV-2 and another in which they use IMQ linked to a lipid to facilitate its entry into cell membranes

Comments 11: Considering the important role of TLR7 signal in potentiating immune response authors do not discuss much the fact, that, in the human population, there are variants of TLR7 that could jeopardize the efficiency of IMQ therapy/adjuvant. In addition, TLR8 that plays a role in the regulation of TLR7 has also been ignored in the discussion. 

Response 11: Thank you for pointing this out. We agree with this comment. Therefore, A chapter is added briefly describing these TLRs. Chapter 4, pages 5 and 6.

5. Additional clarifications

1. Summary

Response and Revisions: We give corresponding response in the point-by-point response letter. The same as below.

General Comments: Comments and Suggestions for Authors:

The review “Immune stimulation with imiquimod to best face SARS-CoV-2 infection and prevent long COVID” by Ursino Pacheco-Gracia and colleagues develop the hypothesis of a possible therapy to prevent, or reduce, long COVID, a sequel of COVID. The proposed strategy is to associate imiquimod (IMQ), to enhance the immune response, to SARS-Cov2 vaccination in individuals that are potentially low responders to vaccination. Moreover, authors claim for a beneficial use of IMQ during COVID as means of reducing the risk of long COVID.

The manuscript often lacks a connection between the paragraphs (for example line 68-71), and seldom is repetitive. The manuscript requires considerable revision to be accepted for publication.

Response:

Thank you for pointing this out. We agree with this comment.

In the manuscript, we suggest that IMQ could increase the efficiency of both innate and adaptive immunity to better confront potential SARS-CoV-2 infections and reinfections, mainly in individuals with dysregulated immunity due to being of adult age or for suffering from risk factors such as metabolic syndrome (obesity, hypertension, diabetes, senescence, among others), as well as to develop better immunity induced by the application of vaccines against this coronavirus. Individuals with a stimulated immune system could control the infection in less time and prevent the development of systemic infections and organ damage, thus preventing LC.

This also makes it clear that in individuals with deficiencies in TLR7 and TLR8, this treatment would not induce this immunostimulation.

Points to be addressed:

Comments 1: In line 63 authors partially describe changes occurring in the immune function with age, why to emphasise only T cells?

Response 1: Thank you for pointing this out. We agree with this comment. Therefore, in the corrected manuscript, senescence is described more fully in this same paragraph, in lines 59 to 66.

Comments 2:  In line 112 authors attribute a higher risk to develop long COVID to “the presence of FoxP4 locus” in certain subjects, this requires a bit more of explanation because maybe is not the locus but the expression of a certain allele/gene variants in that locus.

Response 2: Thank you for pointing this out. We agree with this comment. Therefore, The information is expanded at this point, in lines 106 to 115.

Comments 3: In line 120 the authors start to speak of organ damage, they refer the importance of targeting the thymus but then they move to infant COVID infections and is not clear which is the connection. And if it is the thymic dysfunction how if may affect the response to other infections/vaccinations.

Response 3: Thank you for pointing this out. We agree with this comment. Therefore, information about the topic is added and separated from the following text, in lines 127 to 129, page 4.

Comments 4: In line 162 the authors start a chapter in which they revise the literature on the benefits/ damage of the use IMQ. I would suggest the authors to start to describe what is this molecule, line 212, also to avoid the repetition shown in the beginning of chapter 5.

Response 4: Thank you for pointing this out. We agree with this comment. Therefore, the information that was in the following chapter is adapted and added here. Chapter 5, lines 256-261.

Comments 5: Figure 1 is not of high quality thus numbers on the axis are not visible and colour codes are very similar leading to a bit of confusion.

Response 5: Thank you for pointing this out. We agree with this comment. Therefore, This figure is modified to have a better visualization.  Page 8.

Comments 6: In line 251 consider to add the abbreviation for human papilloma virus because later on is used in the text.

Response 6: Thank you for pointing this out. We agree with this comment. Therefore, This change has been made, Page 9, line 349.

Comments 7: In line 253 authors state that IMQ activates 3000 genes, repeating what was previously described for cytokine secretion, etc, etc. few lines before; maybe is worth to better integrate this part of the text

Response 7: Thank you for pointing this out. We agree with this comment. Therefore, The text is adapted as suggested.lines 344-346, Page 9.

Comments 8: In line 285 authors describe the use of IMQ in disease conditions in which it has been approved by FDA and EMA. Of note that these infections are quite localized and authors do not discuss how IMQ can really be used in patients with COVID

Response 8: Thank you for pointing this out. We agree with this comment. It is indicated that topical application of this medication has systemic effects and stimulates general immunity including mucosal immunity. Lines 385-387, page 11.

Comments 9: In line 311 please make the spelling of VPH.

Response 9: Thank you for pointing this out. We agree with this comment. Therefore, This suggestion has been made.line 408, page 12.

Comments 10: In line 326 authors refer to a study in which IMQ has been used to potentiate response to vaccination. Maybe is useful to give a reference for the work done on COVID, view that authors devote the following lines to describe response to anti-influenza vaccination in mouse models.

Response 10: Thank you for pointing this out. We agree with this comment. In lines 444-463, pge 13, onwards, two reported works on vaccines against SARS-CoV-2 are discussed in which they use IMQ linked to the S1 protein of SARS-CoV-2 and another in which they use IMQ linked to a lipid to facilitate its entry into cell membranes

Comments 11: Considering the important role of TLR7 signal in potentiating immune response authors do not discuss much the fact, that, in the human population, there are variants of TLR7 that could jeopardize the efficiency of IMQ therapy/adjuvant. In addition, TLR8 that plays a role in the regulation of TLR7 has also been ignored in the discussion. 

Response 11: Thank you for pointing this out. We agree with this comment. Therefore, A chapter is added briefly describing these TLRs. Chapter 4, pages 5 and 6.

5. Additional clarifications

1. Summary

Response and Revisions: We give corresponding response in the point-by-point response letter. The same as below.

General Comments: Comments and Suggestions for Authors:

The review “Immune stimulation with imiquimod to best face SARS-CoV-2 infection and prevent long COVID” by Ursino Pacheco-Gracia and colleagues develop the hypothesis of a possible therapy to prevent, or reduce, long COVID, a sequel of COVID. The proposed strategy is to associate imiquimod (IMQ), to enhance the immune response, to SARS-Cov2 vaccination in individuals that are potentially low responders to vaccination. Moreover, authors claim for a beneficial use of IMQ during COVID as means of reducing the risk of long COVID.

The manuscript often lacks a connection between the paragraphs (for example line 68-71), and seldom is repetitive. The manuscript requires considerable revision to be accepted for publication.

Response:

Thank you for pointing this out. We agree with this comment.

In the manuscript, we suggest that IMQ could increase the efficiency of both innate and adaptive immunity to better confront potential SARS-CoV-2 infections and reinfections, mainly in individuals with dysregulated immunity due to being of adult age or for suffering from risk factors such as metabolic syndrome (obesity, hypertension, diabetes, senescence, among others), as well as to develop better immunity induced by the application of vaccines against this coronavirus. Individuals with a stimulated immune system could control the infection in less time and prevent the development of systemic infections and organ damage, thus preventing LC.

This also makes it clear that in individuals with deficiencies in TLR7 and TLR8, this treatment would not induce this immunostimulation.

Points to be addressed:

Comments 1: In line 63 authors partially describe changes occurring in the immune function with age, why to emphasise only T cells?

Response 1: Thank you for pointing this out. We agree with this comment. Therefore, in the corrected manuscript, senescence is described more fully in this same paragraph, in lines 59 to 66.

Comments 2:  In line 112 authors attribute a higher risk to develop long COVID to “the presence of FoxP4 locus” in certain subjects, this requires a bit more of explanation because maybe is not the locus but the expression of a certain allele/gene variants in that locus.

Response 2: Thank you for pointing this out. We agree with this comment. Therefore, The information is expanded at this point, in lines 106 to 115.

Comments 3: In line 120 the authors start to speak of organ damage, they refer the importance of targeting the thymus but then they move to infant COVID infections and is not clear which is the connection. And if it is the thymic dysfunction how if may affect the response to other infections/vaccinations.

Response 3: Thank you for pointing this out. We agree with this comment. Therefore, information about the topic is added and separated from the following text, in lines 127 to 129, page 4.

Comments 4: In line 162 the authors start a chapter in which they revise the literature on the benefits/ damage of the use IMQ. I would suggest the authors to start to describe what is this molecule, line 212, also to avoid the repetition shown in the beginning of chapter 5.

Response 4: Thank you for pointing this out. We agree with this comment. Therefore, the information that was in the following chapter is adapted and added here. Chapter 5, lines 256-261.

Comments 5: Figure 1 is not of high quality thus numbers on the axis are not visible and colour codes are very similar leading to a bit of confusion.

Response 5: Thank you for pointing this out. We agree with this comment. Therefore, This figure is modified to have a better visualization.  Page 8.

Comments 6: In line 251 consider to add the abbreviation for human papilloma virus because later on is used in the text.

Response 6: Thank you for pointing this out. We agree with this comment. Therefore, This change has been made, Page 9, line 349.

Comments 7: In line 253 authors state that IMQ activates 3000 genes, repeating what was previously described for cytokine secretion, etc, etc. few lines before; maybe is worth to better integrate this part of the text

Response 7: Thank you for pointing this out. We agree with this comment. Therefore, The text is adapted as suggested.lines 344-346, Page 9.

Comments 8: In line 285 authors describe the use of IMQ in disease conditions in which it has been approved by FDA and EMA. Of note that these infections are quite localized and authors do not discuss how IMQ can really be used in patients with COVID

Response 8: Thank you for pointing this out. We agree with this comment. It is indicated that topical application of this medication has systemic effects and stimulates general immunity including mucosal immunity. Lines 385-387, page 11.

Comments 9: In line 311 please make the spelling of VPH.

Response 9: Thank you for pointing this out. We agree with this comment. Therefore, This suggestion has been made.line 408, page 12.

Comments 10: In line 326 authors refer to a study in which IMQ has been used to potentiate response to vaccination. Maybe is useful to give a reference for the work done on COVID, view that authors devote the following lines to describe response to anti-influenza vaccination in mouse models.

Response 10: Thank you for pointing this out. We agree with this comment. In lines 444-463, pge 13, onwards, two reported works on vaccines against SARS-CoV-2 are discussed in which they use IMQ linked to the S1 protein of SARS-CoV-2 and another in which they use IMQ linked to a lipid to facilitate its entry into cell membranes

Comments 11: Considering the important role of TLR7 signal in potentiating immune response authors do not discuss much the fact, that, in the human population, there are variants of TLR7 that could jeopardize the efficiency of IMQ therapy/adjuvant. In addition, TLR8 that plays a role in the regulation of TLR7 has also been ignored in the discussion. 

Response 11: Thank you for pointing this out. We agree with this comment. Therefore, A chapter is added briefly describing these TLRs. Chapter 4, pages 5 and 6.

5. Additional clarifications

1. Summary

Response and Revisions: We give corresponding response in the point-by-point response letter. The same as below.

General Comments: Comments and Suggestions for Authors:

The review “Immune stimulation with imiquimod to best face SARS-CoV-2 infection and prevent long COVID” by Ursino Pacheco-Gracia and colleagues develop the hypothesis of a possible therapy to prevent, or reduce, long COVID, a sequel of COVID. The proposed strategy is to associate imiquimod (IMQ), to enhance the immune response, to SARS-Cov2 vaccination in individuals that are potentially low responders to vaccination. Moreover, authors claim for a beneficial use of IMQ during COVID as means of reducing the risk of long COVID.

The manuscript often lacks a connection between the paragraphs (for example line 68-71), and seldom is repetitive. The manuscript requires considerable revision to be accepted for publication.

Response:

Thank you for pointing this out. We agree with this comment.

In the manuscript, we suggest that IMQ could increase the efficiency of both innate and adaptive immunity to better confront potential SARS-CoV-2 infections and reinfections, mainly in individuals with dysregulated immunity due to being of adult age or for suffering from risk factors such as metabolic syndrome (obesity, hypertension, diabetes, senescence, among others), as well as to develop better immunity induced by the application of vaccines against this coronavirus. Individuals with a stimulated immune system could control the infection in less time and prevent the development of systemic infections and organ damage, thus preventing LC.

This also makes it clear that in individuals with deficiencies in TLR7 and TLR8, this treatment would not induce this immunostimulation.

Points to be addressed:

Comments 1: In line 63 authors partially describe changes occurring in the immune function with age, why to emphasise only T cells?

Response 1: Thank you for pointing this out. We agree with this comment. Therefore, in the corrected manuscript, senescence is described more fully in this same paragraph, in lines 59 to 66.

Comments 2:  In line 112 authors attribute a higher risk to develop long COVID to “the presence of FoxP4 locus” in certain subjects, this requires a bit more of explanation because maybe is not the locus but the expression of a certain allele/gene variants in that locus.

Response 2: Thank you for pointing this out. We agree with this comment. Therefore, The information is expanded at this point, in lines 106 to 115.

Comments 3: In line 120 the authors start to speak of organ damage, they refer the importance of targeting the thymus but then they move to infant COVID infections and is not clear which is the connection. And if it is the thymic dysfunction how if may affect the response to other infections/vaccinations.

Response 3: Thank you for pointing this out. We agree with this comment. Therefore, information about the topic is added and separated from the following text, in lines 127 to 129, page 4.

Comments 4: In line 162 the authors start a chapter in which they revise the literature on the benefits/ damage of the use IMQ. I would suggest the authors to start to describe what is this molecule, line 212, also to avoid the repetition shown in the beginning of chapter 5.

Response 4: Thank you for pointing this out. We agree with this comment. Therefore, the information that was in the following chapter is adapted and added here. Chapter 5, lines 256-261.

Comments 5: Figure 1 is not of high quality thus numbers on the axis are not visible and colour codes are very similar leading to a bit of confusion.

Response 5: Thank you for pointing this out. We agree with this comment. Therefore, This figure is modified to have a better visualization.  Page 8.

Comments 6: In line 251 consider to add the abbreviation for human papilloma virus because later on is used in the text.

Response 6: Thank you for pointing this out. We agree with this comment. Therefore, This change has been made, Page 9, line 349.

Comments 7: In line 253 authors state that IMQ activates 3000 genes, repeating what was previously described for cytokine secretion, etc, etc. few lines before; maybe is worth to better integrate this part of the text

Response 7: Thank you for pointing this out. We agree with this comment. Therefore, The text is adapted as suggested.lines 344-346, Page 9.

Comments 8: In line 285 authors describe the use of IMQ in disease conditions in which it has been approved by FDA and EMA. Of note that these infections are quite localized and authors do not discuss how IMQ can really be used in patients with COVID

Response 8: Thank you for pointing this out. We agree with this comment. It is indicated that topical application of this medication has systemic effects and stimulates general immunity including mucosal immunity. Lines 385-387, page 11.

Comments 9: In line 311 please make the spelling of VPH.

Response 9: Thank you for pointing this out. We agree with this comment. Therefore, This suggestion has been made.line 408, page 12.

Comments 10: In line 326 authors refer to a study in which IMQ has been used to potentiate response to vaccination. Maybe is useful to give a reference for the work done on COVID, view that authors devote the following lines to describe response to anti-influenza vaccination in mouse models.

Response 10: Thank you for pointing this out. We agree with this comment. In lines 444-463, pge 13, onwards, two reported works on vaccines against SARS-CoV-2 are discussed in which they use IMQ linked to the S1 protein of SARS-CoV-2 and another in which they use IMQ linked to a lipid to facilitate its entry into cell membranes

Comments 11: Considering the important role of TLR7 signal in potentiating immune response authors do not discuss much the fact, that, in the human population, there are variants of TLR7 that could jeopardize the efficiency of IMQ therapy/adjuvant. In addition, TLR8 that plays a role in the regulation of TLR7 has also been ignored in the discussion. 

Response 11: Thank you for pointing this out. We agree with this comment. Therefore, A chapter is added briefly describing these TLRs. Chapter 4, pages 5 and 6.

5. Additional clarifications

1. Summary

Response and Revisions: We give corresponding response in the point-by-point response letter. The same as below.

General Comments: Comments and Suggestions for Authors:

The review “Immune stimulation with imiquimod to best face SARS-CoV-2 infection and prevent long COVID” by Ursino Pacheco-Gracia and colleagues develop the hypothesis of a possible therapy to prevent, or reduce, long COVID, a sequel of COVID. The proposed strategy is to associate imiquimod (IMQ), to enhance the immune response, to SARS-Cov2 vaccination in individuals that are potentially low responders to vaccination. Moreover, authors claim for a beneficial use of IMQ during COVID as means of reducing the risk of long COVID.

The manuscript often lacks a connection between the paragraphs (for example line 68-71), and seldom is repetitive. The manuscript requires considerable revision to be accepted for publication.

Response:

Thank you for pointing this out. We agree with this comment.

In the manuscript, we suggest that IMQ could increase the efficiency of both innate and adaptive immunity to better confront potential SARS-CoV-2 infections and reinfections, mainly in individuals with dysregulated immunity due to being of adult age or for suffering from risk factors such as metabolic syndrome (obesity, hypertension, diabetes, senescence, among others), as well as to develop better immunity induced by the application of vaccines against this coronavirus. Individuals with a stimulated immune system could control the infection in less time and prevent the development of systemic infections and organ damage, thus preventing LC.

This also makes it clear that in individuals with deficiencies in TLR7 and TLR8, this treatment would not induce this immunostimulation.

Points to be addressed:

Comments 1: In line 63 authors partially describe changes occurring in the immune function with age, why to emphasise only T cells?

Response 1: Thank you for pointing this out. We agree with this comment. Therefore, in the corrected manuscript, senescence is described more fully in this same paragraph, in lines 59 to 66.

Comments 2:  In line 112 authors attribute a higher risk to develop long COVID to “the presence of FoxP4 locus” in certain subjects, this requires a bit more of explanation because maybe is not the locus but the expression of a certain allele/gene variants in that locus.

Response 2: Thank you for pointing this out. We agree with this comment. Therefore, The information is expanded at this point, in lines 106 to 115.

Comments 3: In line 120 the authors start to speak of organ damage, they refer the importance of targeting the thymus but then they move to infant COVID infections and is not clear which is the connection. And if it is the thymic dysfunction how if may affect the response to other infections/vaccinations.

Response 3: Thank you for pointing this out. We agree with this comment. Therefore, information about the topic is added and separated from the following text, in lines 127 to 129, page 4.

Comments 4: In line 162 the authors start a chapter in which they revise the literature on the benefits/ damage of the use IMQ. I would suggest the authors to start to describe what is this molecule, line 212, also to avoid the repetition shown in the beginning of chapter 5.

Response 4: Thank you for pointing this out. We agree with this comment. Therefore, the information that was in the following chapter is adapted and added here. Chapter 5, lines 256-261.

Comments 5: Figure 1 is not of high quality thus numbers on the axis are not visible and colour codes are very similar leading to a bit of confusion.

Response 5: Thank you for pointing this out. We agree with this comment. Therefore, This figure is modified to have a better visualization.  Page 8.

Comments 6: In line 251 consider to add the abbreviation for human papilloma virus because later on is used in the text.

Response 6: Thank you for pointing this out. We agree with this comment. Therefore, This change has been made, Page 9, line 349.

Comments 7: In line 253 authors state that IMQ activates 3000 genes, repeating what was previously described for cytokine secretion, etc, etc. few lines before; maybe is worth to better integrate this part of the text

Response 7: Thank you for pointing this out. We agree with this comment. Therefore, The text is adapted as suggested.lines 344-346, Page 9.

Comments 8: In line 285 authors describe the use of IMQ in disease conditions in which it has been approved by FDA and EMA. Of note that these infections are quite localized and authors do not discuss how IMQ can really be used in patients with COVID

Response 8: Thank you for pointing this out. We agree with this comment. It is indicated that topical application of this medication has systemic effects and stimulates general immunity including mucosal immunity. Lines 385-387, page 11.

Comments 9: In line 311 please make the spelling of VPH.

Response 9: Thank you for pointing this out. We agree with this comment. Therefore, This suggestion has been made.line 408, page 12.

Comments 10: In line 326 authors refer to a study in which IMQ has been used to potentiate response to vaccination. Maybe is useful to give a reference for the work done on COVID, view that authors devote the following lines to describe response to anti-influenza vaccination in mouse models.

Response 10: Thank you for pointing this out. We agree with this comment. In lines 444-463, pge 13, onwards, two reported works on vaccines against SARS-CoV-2 are discussed in which they use IMQ linked to the S1 protein of SARS-CoV-2 and another in which they use IMQ linked to a lipid to facilitate its entry into cell membranes

Comments 11: Considering the important role of TLR7 signal in potentiating immune response authors do not discuss much the fact, that, in the human population, there are variants of TLR7 that could jeopardize the efficiency of IMQ therapy/adjuvant. In addition, TLR8 that plays a role in the regulation of TLR7 has also been ignored in the discussion. 

Response 11: Thank you for pointing this out. We agree with this comment. Therefore, A chapter is added briefly describing these TLRs. Chapter 4, pages 5 and 6.

5. Additional clarifications

1. Summary

Response and Revisions: We give corresponding response in the point-by-point response letter. The same as below.

General Comments: Comments and Suggestions for Authors:

The review “Immune stimulation with imiquimod to best face SARS-CoV-2 infection and prevent long COVID” by Ursino Pacheco-Gracia and colleagues develop the hypothesis of a possible therapy to prevent, or reduce, long COVID, a sequel of COVID. The proposed strategy is to associate imiquimod (IMQ), to enhance the immune response, to SARS-Cov2 vaccination in individuals that are potentially low responders to vaccination. Moreover, authors claim for a beneficial use of IMQ during COVID as means of reducing the risk of long COVID.

The manuscript often lacks a connection between the paragraphs (for example line 68-71), and seldom is repetitive. The manuscript requires considerable revision to be accepted for publication.

Response:

Thank you for pointing this out. We agree with this comment.

In the manuscript, we suggest that IMQ could increase the efficiency of both innate and adaptive immunity to better confront potential SARS-CoV-2 infections and reinfections, mainly in individuals with dysregulated immunity due to being of adult age or for suffering from risk factors such as metabolic syndrome (obesity, hypertension, diabetes, senescence, among others), as well as to develop better immunity induced by the application of vaccines against this coronavirus. Individuals with a stimulated immune system could control the infection in less time and prevent the development of systemic infections and organ damage, thus preventing LC.

This also makes it clear that in individuals with deficiencies in TLR7 and TLR8, this treatment would not induce this immunostimulation.

Points to be addressed:

Comments 1: In line 63 authors partially describe changes occurring in the immune function with age, why to emphasise only T cells?

Response 1: Thank you for pointing this out. We agree with this comment. Therefore, in the corrected manuscript, senescence is described more fully in this same paragraph, in lines 59 to 66.

Comments 2:  In line 112 authors attribute a higher risk to develop long COVID to “the presence of FoxP4 locus” in certain subjects, this requires a bit more of explanation because maybe is not the locus but the expression of a certain allele/gene variants in that locus.

Response 2: Thank you for pointing this out. We agree with this comment. Therefore, The information is expanded at this point, in lines 106 to 115.

Comments 3: In line 120 the authors start to speak of organ damage, they refer the importance of targeting the thymus but then they move to infant COVID infections and is not clear which is the connection. And if it is the thymic dysfunction how if may affect the response to other infections/vaccinations.

Response 3: Thank you for pointing this out. We agree with this comment. Therefore, information about the topic is added and separated from the following text, in lines 127 to 129, page 4.

Comments 4: In line 162 the authors start a chapter in which they revise the literature on the benefits/ damage of the use IMQ. I would suggest the authors to start to describe what is this molecule, line 212, also to avoid the repetition shown in the beginning of chapter 5.

Response 4: Thank you for pointing this out. We agree with this comment. Therefore, the information that was in the following chapter is adapted and added here. Chapter 5, lines 256-261.

Comments 5: Figure 1 is not of high quality thus numbers on the axis are not visible and colour codes are very similar leading to a bit of confusion.

Response 5: Thank you for pointing this out. We agree with this comment. Therefore, This figure is modified to have a better visualization.  Page 8.

Comments 6: In line 251 consider to add the abbreviation for human papilloma virus because later on is used in the text.

Response 6: Thank you for pointing this out. We agree with this comment. Therefore, This change has been made, Page 9, line 349.

Comments 7: In line 253 authors state that IMQ activates 3000 genes, repeating what was previously described for cytokine secretion, etc, etc. few lines before; maybe is worth to better integrate this part of the text

Response 7: Thank you for pointing this out. We agree with this comment. Therefore, The text is adapted as suggested.lines 344-346, Page 9.

Comments 8: In line 285 authors describe the use of IMQ in disease conditions in which it has been approved by FDA and EMA. Of note that these infections are quite localized and authors do not discuss how IMQ can really be used in patients with COVID

Response 8: Thank you for pointing this out. We agree with this comment. It is indicated that topical application of this medication has systemic effects and stimulates general immunity including mucosal immunity. Lines 385-387, page 11.

Comments 9: In line 311 please make the spelling of VPH.

Response 9: Thank you for pointing this out. We agree with this comment. Therefore, This suggestion has been made.line 408, page 12.

Comments 10: In line 326 authors refer to a study in which IMQ has been used to potentiate response to vaccination. Maybe is useful to give a reference for the work done on COVID, view that authors devote the following lines to describe response to anti-influenza vaccination in mouse models.

Response 10: Thank you for pointing this out. We agree with this comment. In lines 444-463, pge 13, onwards, two reported works on vaccines against SARS-CoV-2 are discussed in which they use IMQ linked to the S1 protein of SARS-CoV-2 and another in which they use IMQ linked to a lipid to facilitate its entry into cell membranes

Comments 11: Considering the important role of TLR7 signal in potentiating immune response authors do not discuss much the fact, that, in the human population, there are variants of TLR7 that could jeopardize the efficiency of IMQ therapy/adjuvant. In addition, TLR8 that plays a role in the regulation of TLR7 has also been ignored in the discussion. 

Response 11: Thank you for pointing this out. We agree with this comment. Therefore, A chapter is added briefly describing these TLRs. Chapter 4, pages 5 and 6.

5. Additional clarifications

Reviewer 2 Report

Comments and Suggestions for Authors

The review examined an immunostimulatory treatment regimen using imiquimod (IMQ), which has the potential to positively modify and enhance the immune response in people with dysregulated immune systems.

1. Please shorten the introduction that is not related to the issue discussed.

2. Please provide the doses of imiquimod used throughout the article

Comments on the Quality of English Language

Minor editing of English language required

Author Response

1. Summary

Response and Revisions: We give corresponding response in the point-by-point response letter. The same as below.

General Comments: Comments and Suggestions for Authors:

The review examined an immunostimulatory treatment regimen using imiquimod (IMQ), which has the potential to positively modify and enhance the immune response in people with dysregulated immune systems.

Response:

Thank you for pointing this out. We agree with this comment.

Points to be addressed:

Comments 1: Please shorten the introduction that is not related to the issue discussed.

Response 1:

Thank you for pointing this out. We agree with this comment. Therefore, we have shortened the introduction although minimally.

Comments 2: Please provide the doses of imiquimod used throughout the article

Response 2:

Thank you for pointing this out. About with this comment, the doses and treatment schedules used in patients with HPV are described in page 10, lines 358; while a suggested application scheme for clinical research protocols is found in Table 1 page 14.   

Additional clarifications

1. Summary

Response and Revisions: We give corresponding response in the point-by-point response letter. The same as below.

General Comments: Comments and Suggestions for Authors:

The review examined an immunostimulatory treatment regimen using imiquimod (IMQ), which has the potential to positively modify and enhance the immune response in people with dysregulated immune systems.

Response:

Thank you for pointing this out. We agree with this comment.

Points to be addressed:

Comments 1: Please shorten the introduction that is not related to the issue discussed.

Response 1:

Thank you for pointing this out. We agree with this comment. Therefore, we have shortened the introduction although minimally.

Comments 2: Please provide the doses of imiquimod used throughout the article

Response 2:

Thank you for pointing this out. About with this comment, the doses and treatment schedules used in patients with HPV are described in page 10, lines 358; while a suggested application scheme for clinical research protocols is found in Table 1 page 14.   

Additional clarifications

1. Summary

Response and Revisions: We give corresponding response in the point-by-point response letter. The same as below.

General Comments: Comments and Suggestions for Authors:

The review examined an immunostimulatory treatment regimen using imiquimod (IMQ), which has the potential to positively modify and enhance the immune response in people with dysregulated immune systems.

Response:

Thank you for pointing this out. We agree with this comment.

Points to be addressed:

Comments 1: Please shorten the introduction that is not related to the issue discussed.

Response 1:

Thank you for pointing this out. We agree with this comment. Therefore, we have shortened the introduction although minimally.

Comments 2: Please provide the doses of imiquimod used throughout the article

Response 2:

Thank you for pointing this out. About with this comment, the doses and treatment schedules used in patients with HPV are described in page 10, lines 358; while a suggested application scheme for clinical research protocols is found in Table 1 page 14.   

Additional clarifications

Round 2

Reviewer 1 Report

Comments and Suggestions for Authors

Authors answered all the raise critical points